# Quality-Driven Curation of Remote Sensing Vision-Language Data via Learned Scoring Models

**Dilxat Muhtar**      **Enzhuo Zhang**      **Zhenshi Li**      **Feng Gu**      **Yanglangxing He**

**Pengfeng Xiao**      **Xueliang Zhang**[*]

**Nanjing University**

pumpkindilxat@gmail.com, Zenzhuo@smail.nju.edu.cn
{xiaopf, zxl}@nju.edu.cn

## Abstract

Vision-Language Models (VLMs) have demonstrated great potential in interpreting remote sensing (RS) images through language-guided semantic. However, the effectiveness of these VLMs critically depends on high-quality image-text training data that captures rich semantic relationships between visual content and language descriptions. Unlike natural images, RS lacks large-scale interleaved image-text pairs from web data, making data collection challenging. While current approaches rely primarily on rule-based methods or flagship VLMs for data synthesis, a systematic framework for automated quality assessment of such synthetically generated RS vision-language data is notably absent. To fill this gap, we propose a novel score model trained on large-scale RS vision-language preference data for automated quality assessment. Our empirical results demonstrate that fine-tuning CLIP or advanced VLMs (e.g., Qwen2-VL) with the top 30% of data ranked by our score model achieves superior accuracy compared to both full-data fine-tuning and CLIP-score-based ranking approaches. Furthermore, we demonstrate applications of our scoring model for reinforcement learning (RL) training and best-of-N (BoN) test-time scaling, enabling significant improvements in VLM performance for RS tasks. Our code, model, and dataset are publicly available[2].

## 1 Introduction

The advancement of artificial intelligence has consistently benefited from scaling across three crucial dimensions: data volume, computational resources, and model complexity [26, 21]. In visual understanding, vision-language models (VLMs) have particularly benefited from diverse training data, exhibiting evolutionary patterns that mirror human cognitive development in perception and interpretation. From the foundational CLIP model [46] to recent integrations with large language models (LLMs) [8, 61, 32, 15], VLMs have achieved remarkable success, emerging as indispensable tools for real-world applications ranging from agents system [81] to autonomous driving [43].

Despite these advancements, VLMs demonstrate limited capability in interpreting remote sensing (RS) images due to substantial domain shifts in data distributions [31, 27]. This performance gap stems primarily from a critical bottleneck: the scarcity of large-scale vision-language data for RS domains, which, unlike natural images, lacks the abundant, naturally aligned image-text pairs readily

---

[*]Corresponding Author
[2] 🤗 Data    🤗 Model    ⓞ Code

39th Conference on Neural Information Processing Systems (NeurIPS 2025).

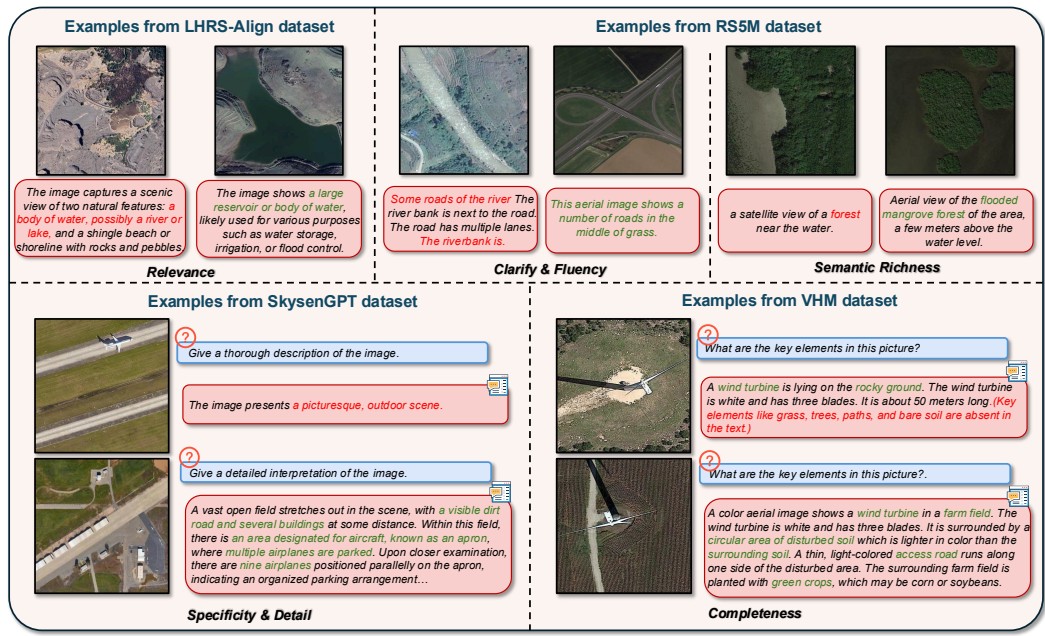

Figure 1: Examples from RS vision-language datasets showing quality issues across five dimensions. Green represents reasonably good expression, while red represents low-quality expression

available through web crawling. Current approaches to bridge this gap include: (1) binary classification models filtering RS data from general vision-language datasets like DataComp [79], (2) rule-based construction of vision-language pairs using OpenStreetMap tags [62], and (3) automated data generation through flagship VLMs [36, 27, 39, 30, 44] or leveraging supervision from internet images [38]. While these methods advance holistic RS understanding, they introduce significant quality concerns: rule-based approaches often produce informationally sparse or semantically inconsistent pairs, while VLM-generated data risks propagating hallucinatory content that misrepresents the actual imagery and incorrectly infers answers to questions that cannot be determined from the given images (Figure 1). The absence of robust, automated quality assessment framework consequently constrains further improvements in RS-specific VLMs [30]

To address this critical gap, our method commences with the definition of text quality for RS images, outlining five crucial dimensions that characterize high-quality descriptions or instruction samples: *relevance* to visual content [44, 36, 70], *specificity and detail* [30, 29, 83], *completeness* of coverage for salient features [4, 83], *clarity and fluency* of language [62], and *semantic richness* for RS applications [63, 52]. Guided by these quality dimensions, we construct preference datasets for both image-captions and vision instructions by employing a diverse array of policy models—including RS-specialized VLMs and open-source alternatives—while utilizing a combination of rule-based evaluation metrics and flagship VLMs as objective judges. Building upon these curated preference datasets, we introduce ScoreRS, a novel learned quality scoring model tailored for RS vision-language data, developed via a three-stage progressive training recipe. We evaluate ScoreRS's effectiveness in both data quality ranking and its practical applications as a large VLM reward model for group relative policy optimization (GRPO) [51] reinforcement learning (RL) and best-of-n (BoN) selector. Our results demonstrate that fine-tuning CLIP and Qwen2VL [61] on just 30% of the data, ranked using ScoreRS scores, outperforms models trained on either the complete dataset or CLIP-score ranked data. Furthermore, evaluation on the challenging RS-specific vision-language benchmark VG-DIOR [74] and LHRS-Bench [39] reveals that ScoreRS can be integrated with rule-based rewards for GRPO RL to enhance model capabilities, and serves as an effective BoN selector for improving results when scaling VLMs with multiple generated samples at test time.

The main contributions of our work can be summarized as follows:

1. We establish a framework for defining text quality in RS vision-language data and introduce the first large-scale RS-specific preference dataset, comprising pairwise preference pairs for both image-captions and vision instructions. Based on this dataset, we develop ScoreRS—a

novel data scoring model specifically designed for automated quality assessment of RS vision-language data.

2. We demonstrate that models fine-tuned on just the top 30% of RS vision-language data ranked by ScoreRS scores consistently outperform those trained on either the complete dataset or data ranked using CLIP scores, highlighting the efficacy of our quality-driven curation approach.

3. We validate ScoreRS's effectiveness through two practical applications: (1) as a reward model for GRPO-based RL, and (2) as a BoN selector for VLM-generated samples at test time. Both applications yield improvements in RS VLM performance across multiple challenging benchmarks.

## 2 Related Work

### 2.1 RS Vision-Language Data Curation

Unlike the general vision domain, where vision-language data can be readily crawled from abundant image-text interleaved webpages [69, 16, 57, 48, 7], the RS domain presents a challenge. While rich in open-source image data [39, 40, 54], RS lacks corresponding text descriptions and analytical conversations about these images. To harness the potential of VLMs in the RS domain, existing studies have primarily focused on two approaches for data curation: rule-based methods and synthetic data generation using established VLMs. For rule-based approaches, RS5M [79] explores the use of binary RS image classifiers to select RS image-text pairs from open-source general vision-language datasets. Similarly, RemoteCLIP [31], Skyscript [62], and SkysenseGPT [36] implement manually designed text templates or relation graphs for data construction. In parallel, studies such as VHM [44], LHRS-Bot [39], LHRS-Bot-Nova [30], and GeoChat [27] leverage flagship VLMs like GPT4 or Gemini for synthetic vision-language data curation. While these works have advanced the development of RS-specific VLMs, our practical applications reveal that data quality remains suboptimal, motivating the development of automated data selection methods.

### 2.2 RS VLMs

The remarkable success of VLMs in understanding images and engaging in complex tasks [35, 43] has inspired the development of specialized VLMs for RS image interpretation. Several approaches have focused on adapting CLIP [46] for the RS domain. RemoteCLIP [31], Skyscript [62], and RS5M [79] fine-tuned CLIP using carefully curated RS image-caption pairs, enabling robust RS zero-shot classification and image-text retrieval capabilities. For larger VLMs, GeoChat [27] pioneered the fine-tuning of LLaVA [32] with RS-specific vision instruction datasets. Subsequent works have expanded this paradigm through various strategies: incorporating volunteer geographic information (VGI) [39], integrating high-quality vision-language data [30, 44], modeling relational graphs [36], implementing ensembled vision encoders for vision-centric designs [78], improving temporal understanding of RS images [24], and scaling data volume with significantly smaller encoder-decoder architectures [71]. The versatile and successful applications of RS VLMs have directed our attention to the fundamental building blocks underlying these models: data quality and utilization. Rather than proposing new architectural designs, our work focuses on addressing how to improve data quality and enhance VLM performance within existing frameworks.

### 2.3 Parameter-Wise Data Selection

As data volume continues to expand and data complexity evolves, quality control has become increasingly challenging. Beyond traditional rule-based data selection methods [53, 64], researchers have shifted focus toward parameter-wise selection approaches for more sophisticated data curation. In the language domain, methods such as QaRater [65] and LESS [68] explore training dedicated scoring models or leveraging model gradients to select high-quality data for LLM pretraining and instruction tuning. Similarly, for vision-language data, Zhang et al. [77] investigated scoring models for curating image-text pairs. The model selection approach has been widely adopted in recent state-of-the-art VLMs, including Llama-3.2-Vision [18], DeepseekVL2 [66], and Qwen2VL [61], all of which implement scoring models for data filtering. Despite these advances, there is a notable

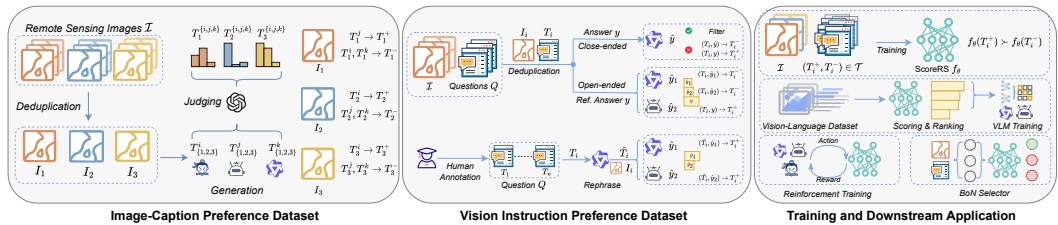

Figure 2: Pipeline for generating pairwise preference datasets and the training/application of our ScoreRS model. $I_i \in \mathcal{I}$ represents a RS image, and $T_i \in \mathcal{T}$ represents an image caption, question, or conversation associated with the image

absence of open-source scoring models specifically designed to evaluate the quality of RS vision-language data. In this work, we develop and release the first open-source scoring model tailored for RS vision-language data quality assessment.

## 3 Method

To establish quality control for RS vision-language datasets, we propose a scoring framework that quantitatively evaluates data quality through a learned model. This framework enables data curation through score-based ranking and selection of high-quality samples.

Formally, we define a scoring function $f_\theta : \mathcal{T} \times \mathcal{I} \to \mathbb{R}$, parameterized by $\theta$, where $\mathcal{T}$ and $\mathcal{I}$ denote the text and image spaces, respectively. Given an image-text pair $(I, T) \in \mathcal{I} \times \mathcal{T}$, which may consist of either a simple caption or a multi-turn conversation associated with the image, the model outputs a scalar quality score $s = f_\theta(I, T)$. The scoring function is trained to assign higher scores to better-aligned image-text pairs $(I, T)$ through pairwise preference learning. For each image $I$, we collect pairs of text $(T^+, T^-)$ where $T^+$ is preferred over $T^-$. This preference dataset is formally defined as $\mathcal{D} = (I_i, T_i^+, T_i^-)_{i=1}^{|\mathcal{D}|}$. The pairwise preferences are modeled using the Bradley-Terry model [3], which defines the probability of $T^+$ being preferred over $T^-$ given image $I$ as:

$$p(T^+ \succ T^- | I) = \frac{\exp(f_\theta(I, T^+))}{\exp(f_\theta(I, T^+)) + \exp(f_\theta(I, T^-))}. \tag{1}$$

The parameters $\theta$ are optimized by minimizing the empirical negative log-likelihood loss [42, 59]:

$$\mathcal{L}(\theta) = -\mathbb{E}_{(I, T^+, T^-) \sim \mathcal{D}} \log(\sigma(f_\theta(I, T^+) - f_\theta(I, T^-))), \tag{2}$$

where $\sigma(\cdot)$ denotes the sigmoid function.

Given the absence of RS-specific vision-language preference datasets, we begin by establishing a framework that defines five key dimensions of textual quality for RS images. This framework characterizes what constitutes high-quality descriptions and question-answer pairs, providing the foundation for our data collection pipeline to construct both image-caption and vision instruction preference pairs (Figure 2). Subsequently, we detail the construction of our scoring function $f_\theta$ by leveraging pre-trained VLMs and employing a progressive training strategy.

### 3.1 Preference Data Construction

### 3.2 Quality Dimensions for RS Vision-Language Data

To develop an effective scoring function $f_\theta$, we must first establish clear criteria for what constitutes high-quality RS vision-language data. These criteria provide the foundation for constructing our preference dataset and subsequently training our score function.

To this end, we define five critical dimensions that characterize high-quality RS vision-language data: *relevance* [44, 36, 70], *specificity & detail* [30, 29, 83], *completeness* [4, 83], *clarity & fluency* [62], and *semantic richness* [63, 52]. For each dimension, we establish a 5-point scoring system (1=Poor to 5=Excellent) with distinct criteria for both image-caption pairs and instruction samples. Detailed descriptions of these quality dimensions, along with illustrative examples and the complete scoring rubrics, are provided in the Appendix A.

### 3.2.1 Image-Caption Preference Dataset

Following established practices in VLM training [61, 39, 44, 32], we begin by constructing an image-caption preference dataset to train ScoreRS to evaluate caption quality for RS images. Given the geographical variations in RS images [39, 62], we utilize the LHRS-Align dataset [39] as our image source. This dataset contains 1.15M orthorectified RS images from major global urban areas, enabling ScoreRS to learn from diverse geographical contexts worldwide. Prior to generating preferences, we implement a rigorous image deduplication process to ensure image quality and representativeness, resulting in 76K distinctive RS images. The detailed method for this deduplication process is provided in Appendix B.1.

For each deduplicated image, we generated captions using three VLMs: RS-specific LHRS-Bot-Nova [30] and general-purpose Qwen2VL-7B [61] and InternVL-2.5-8B [8]. The rationale for selecting these generation models is detailed in Appendix B.2. We generate captions for each image using same prompts and sampling parameters across all models. The resulting captions are then evaluated by GPT-4o based on our predefined scoring system. For each image, we compute the mean score across five dimensions per caption, designating the highest-scoring as positive ($T^+$) and others as negative ($T^-$). After removing results with parsing errors, we construct a dataset of 72K image-caption preference pairs.

To validate GPT-4o's reliability as a preference judge, we conduct a human evaluation study on 1,000 random pairs. Human experts independently annotate these samples to identify positive and negative captions. The results show 92.6% agreement (926 out of 1,000 samples) between human judgments and GPT-4o's assessments. This high level of agreement validates our choice of GPT-4o as a reliable judge for generating pairwise preferences.

### 3.2.2 Vision Instruction Preference Dataset

Vision instruction data consists of conversations about images [32]. To construct a RS-specific vision instruction preference dataset that enables ScoreRS to evaluate responses to diverse user queries, we prioritize collecting a broad spectrum of question types and conversation scenarios. We aggregate vision instruction data from multiple sources: GeoChat [27] (306K samples), LHRS-Instruct [39] (39.8K samples), and a subset of SkysenseGPT [36] (381K samples) as the source of our vision instruction preference dataset. To address potential redundancy across these datasets, we implement a two-stage similarity-based filtering process, resulting in 112K diverse and non-redundant instruction samples. The detailed method for this deduplication procedure is provided in the Appendix B.1.

After deduplication, we categorize conversations into close-ended questions (with definitive answers) and open-ended questions (without verifiable answers). For close-ended questions, we prompt Qwen2VL-7B to generate an answer $\hat{y}$. When $\hat{y}$ differed from the provided answer $y$, we extract the sample, treating $\hat{y}$ as negative target $T^-$ and $y$ as positive target $T^+$.

For open-ended questions, we use LHRS-Bot-Nova and Qwen2VL-7B to generate answers $\hat{y}_1$ and $\hat{y}_2$, then prompt Qwen2VL-72B to evaluate these along with the source dataset answer $y$ using predefined dimensions. We designate the higher scoring answer as the positive target ($T^+$) and others as negative targets ($T^-$). We do not directly consider the answer from the source dataset as $T^+$ because many of them are generated by older VLMs and are often too short, meaningless, or contain incorrect statements. To manage costs associated with the large volume of data, we employ Qwen2VL-72B rather than GPT-4o as our evaluator. After filtering out parsing errors and applying human-defined quality rules, we curate a dataset of 26K vision instruction preference pairs. The detailed prompts used in this process are provided in Appendix B.3.2.

While collecting question set $Q$, we notice most sources lack RS application-specific questions (e.g., agricultural and disaster analysis). To enhance ScoreRS's domain-specific performance, we create a specialized set of five manually crafted questions for each of 12 expert-defined RS image analysis categories. We then randomly sample 35K RS images from our deduplicated 112K image dataset. For each image, we randomly select a question from the manually designed question set and prompt Qwen2.5-13B to rephrase the question to increasing the diversity. LHRS-Bot-Nova and Qwen2VL-7B then generate answers based on these image-question pairs. Finally, GPT-4o evaluates the generated answers, selecting the higher-scoring one as the positive target $T^+$ and treating others as negative targets $T^-$. This process yield 33K RS-specific vision instruction preference data points

after removing entries with parsing errors. The manually designed questions and categories can be found in Appendix B.3.2.

### 3.3 Training and Application

**Training**  Our ScoreRS model is initialized with Qwen2VL-7B, with the language head replaced by a linear layer to output a scalar score, following the standard value-head-based reward model [59, 14]. We do not consider generative score modeling due to efficiency concerns. Similar to the standard recipe for training large VLMs [61, 32], we implement a multi-stage training procedure to gradually train ScoreRS to distinguish better vision-language pair. In the first stage, we train the newly introduced value head using a pure text preference dataset, UltraFeedback [12], to provide a good initialization for the value head. Then, we unfreeze the ViT and value head and train ScoreRS on our image-caption preference dataset to enable ScoreRS to better understand RS images. Finally, we unfreeze the LLM and conduct full-parameter training with our vision instruction preference dataset and the additional RLHF-V [73] dataset. This training approach enables ScoreRS to effectively identify high-quality outputs and assign higher scores to better responses across diverse RS scenarios.

**Application**  Beyond data selection, we also explore ScoreRS's usage for RL training and BoN selection. We primarily discuss implementing ScoreRS for RL, as its applications to data selection and BoN selection are straightforward. Our RL framework is based on GRPO [51], chosen for its computational efficiency and ease of hyperparameter tuning. RL training methods typically require a reward model to evaluate each action trajectory and update the model parameters to favor outputs with higher scores (i.e., responses more aligned with human preferences). While DeepSeek-R1 [19] demonstrated that rule-based rewards for close-ended questions with verifiable answers are effective for RL training, this approach is insufficient for RS applications. In the RS image understanding domain, most questions are open-ended, such as "Describe the urban development patterns visible in this satellite imagery", requiring nuanced interpretations rather than definitive answers. These questions also usually do not have verifiable answers, as different interpretations can lead to multiple acceptable responses. To address this challenge, we introduce a novel reward method using ScoreRS for evaluating open-ended responses while incorporating rule-based rewards for close-ended questions, creating an approach that handles both question types. Specially, for close-ended questions, we employ a binary reward (0 or 1) based on exact match or intersection over union (IoU) with the ground truth. For open-ended questions, where we have a reference answer $y$ (typically sampled from standard vision instruction datasets), we compute the reward $r$ for a predicted answer $\hat{y}$ using the following formulation:

$$r = \begin{cases} 1 - \exp(-(f_\theta(\hat{y}) - f_\theta(y)) \times \beta), & \text{if } f_\theta(\hat{y}) > f_\theta(y) \\ 0, & \text{otherwise} \end{cases} \tag{3}$$

where $\beta > 0$ is a hyperparameter controlling the reward sharpness. We use this reference-based approach because it accelerates learning. The reason behind this approach is that the reference answer serves as a baseline, allowing the policy model to improve upon it by generating responses that receive higher scores. This method focuses on continuous improvement rather than selecting the least problematic option from multiple suboptimal alternatives that may be worse than the reference answer itself. We give our detailed unsuccessful attempts and our reasoning in AppendixC.1, F.2.

## 4  Experiment

We evaluate ScoreRS's effectiveness across three key applications: (1) vision-language data selection for training VLMs, (2) RL training, and (3) BoN selector. We also analyze the impact of score model's size, initialization, and training strategies, along with our curated preference dataset quality.

### 4.1  Vision-Language Data Selection

#### 4.1.1  CLIP Finetuning

**Experimental Setting**  To validate ScoreRS's effectiveness for data selection, we score and rank training samples from RemoteCLIP [31] to finetune CLIP-ViT-L/14 [46]. We compare against three baselines: (1) CLIP without finetuning, (2) CLIP finetuned on the complete dataset, and (3) CLIP finetuned on CLIP-score filtered data. We provide detailed training hyperparameters in

Table 1: Comparison of finetuned CLIP models on classification tasks. Top-1 (@1) and top-5 (@5) classification accuracies are reported.

| | NWPU@1 | NWPU@5 | EuroSAT@1 | EuroSAT@5 | fMoW@1 | fMoW@5 | AID@1 | AID@5 | SIRI-WHU@1 | SIRI-WHU@5 | WHU-RS19@1 | WHU-RS@5 | Avg.@1 | Avg.@5 |
|---|---|---|---|---|---|---|---|---|---|---|---|---|---|---|
| CLIP | 65.31 | 93.23 | 42.14 | 89.20 | **29.40** | 60.21 | 64.11 | 91.21 | 58.11 | 85.17 | 86.24 | 99.21 | 57.55 | 86.37 |
| RemoteCLIP (ALL) | 65.70 | 93.89 | 42.74 | 86.54 | 18.14 | 44.63 | **86.64** | **99.04** | 72.67 | 96.63 | **95.22** | 99.80 | 63.52 | 86.76 |
| RemoteCLIP (30% w. CLIP-Score) | 78.56 | 97.37 | 62.97 | 98.82 | 26.71 | 58.05 | 83.31 | 98.25 | 74.00 | 98.29 | 94.33 | 99.72 | 69.98 | 91.75 |
| RemoteCLIP (30% w. ScoreRS) | **78.58** | **97.54** | **63.67** | **99.01** | 29.29 | **60.70** | 85.14 | 98.41 | **74.21** | **98.87** | 94.95 | **100.00** | **70.97** | **92.42** |

Table 2: Comparison of finetuned CLIP models on cross-modal retrieval tasks. Text-to-image (T2I) and image-to-text (I2T) performance shown using top-1 recall (R@1) and top-5 recall (R@5)

| | UCM T2I R@1 | UCM T2I R@5 | UCM I2T R@1 | UCM I2T R@5 | RSICD T2I R@1 | RSICD T2I R@5 | RSICD I2T R@1 | RSICD I2T R@5 | Avg. R@1 | Avg. R@5 |
|---|---|---|---|---|---|---|---|---|---|---|
| CLIP | 29.44 | 66.84 | 36.67 | 85.24 | 5.31 | 17.09 | 3.75 | 11.99 | 15.78 | 45.29 |
| RemoteCLIP (ALL) | 37.66 | 80.11 | 56.67 | 87.14 | 12.49 | 35.74 | 9.57 | 24.61 | 25.19 | 56.90 |
| RemoteCLIP (30% w. CLIP-Score) | 44.29 | 81.69 | 56.19 | 87.14 | 13.75 | 38.52 | 8.78 | 23.24 | 26.36 | 57.65 |
| RemoteCLIP (30% w. ScoreRS) | **44.56** | **82.09** | **57.90** | **88.40** | **13.90** | 38.02 | **9.59** | **24.88** | **27.11** | **58.35** |

Appendix D.3 and a detailed description of evaluation methodology, including datasets and metrics, in Appendix D.3.1. Moreover, we also compare using ScoreRS for data filtering with filtering with more advanced VLMs like SigLIP-2 [58] and domain specific VLMs like CLIP-LION-RS [62] in Appendix C.3.3.

**Main Result** We evaluate the finetuned models on classification and retrieval tasks. Results in Table 1 and Table 2 challenge the "more data yields better performance" assumption for RS-specific VLMs. CLIP-score filtering improves performance by 5% (classification) and 1% (retrieval), while our ScoreRS filtering further advances these gains to 7% and 2% respectively, outperforming both the complete dataset and CLIP-score filtering. These findings confirm our hypothesis that current RS vision-language data are suboptimal and require quality control.

We further evaluate different parameter-wise data selection methods across various thresholds (Figure 3). Our analysis reveals task-specific patterns: retrieval tasks show higher sensitivity to noise, with greater improvements under stricter filtering, while classification tasks maintain reasonable performance with more relaxed criteria. Notably, quality filtering consistently improves model performance, aligning with previous research that highlights the crucial role of high-quality data in CLIP-style pretraining [69]. More discussion of these results and further comparison in even more extreme data filtering scenarios can be found in Appendix F.1 and C.3.2.

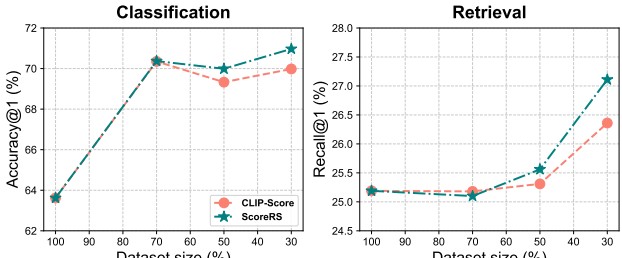

Figure 3: Classification and retrieval results using different percentages of data selected by CLIP-Score and ScoreRS. Top-1 (@1) results shown as average scores across all datasets

### 4.1.2 Large VLMs Finetuning

**Experimental Setting** We fine-tune Qwen2VL-7B [61] using RS image-caption data and vision instruction datasets from VHM [44]. Following standard VLM training practices, we use a two-stage approach: first training only the vision-language bridge layer with pretrained data, then training the LLM with vision instructions. We evaluate ScoreRS's effectiveness in selecting high-quality data for both stages, comparing against models trained on the complete dataset and data filtered by LongCLIP-L [75], which we chose over original CLIP for its superior handling of longer captions and conversations. To reduce computational costs, we implement LoRA [22] with rank 8 for LLM finetuning. Detailed configurations are provided in Appendix D.4.

**Main Result** We evaluate our finetuned model on multiple RS tasks (image classification, visual grounding, question answering) and assess general RS knowledge using the challenging LHRS-Bench [39]. As shown in Table 3, filtering image-caption data with ScoreRS and retaining the top 30% achieves best performance, yielding a 1% improvement on LHRS-Bench. Given that the VHM image-caption dataset is synthetically generated by Gemini-Flash, this high ranking threshold further supports our assertion regarding the importance of quality control for RS synthetic data. We further explore the application of ScoreRS for filtering vision instruction data. Since VHM vision instruction dataset derives from standard RS benchmarks, applying aggressive filtering (e.g., 30%) performs

Table 3: Comparison of different data filtering methods and selection strategies. PX indicates the selection of top X% of image-caption data in the first stage, while SX represents the selection of top X% of vision instruction data in the second stage

| | RS Classification | | | | | | RSVQA | | | | | | Grounding | General Knowledge |
|---|---|---|---|---|---|---|---|---|---|---|---|---|---|---|
| | AID | METERML | NWPU | SIRI-WHU | WHU-RS19 | Avg. | HR-Comp. | HR-Pres. | LR-Comp. | LR-Pres. | LR-R-U | Avg. | VG-DIOR | LHRS-Bench |
| Qwen2VL | 66.13 | 63.54 | 62.35 | 70.79 | 87.40 | 70.04 | 75.60 | 63.30 | 75.47 | 62.00 | 73.00 | 69.87 | 11.87 | 64.78 |
| Qwen2VL-FT | 78.46 | 71.68 | 79.68 | 71.21 | 94.70 | 79.15 | 79.60 | 68.70 | 83.58 | 67.37 | 73.00 | 74.45 | 53.28 | 65.23 |
| Qwen2VL-FT (CLIP-P30) | 78.21 | 72.20 | 78.45 | 62.64 | 94.99 | 77.30 | 82.31 | 67.20 | 84.11 | 69.43 | 73.00 | 75.21 | 54.60 | 65.50 |
| Qwen2VL-FT (ScoreRS-P30) | 79.30 | 72.61 | 80.29 | 72.63 | 95.40 | 80.05 | 85.30 | 70.60 | 85.47 | 68.42 | 76.00 | 77.16 | 55.22 | 66.24 |
| Qwen2VL-FT (CLIP-P30S30) | 76.54 | 65.94 | 77.20 | 60.61 | 90.23 | 74.10 | 78.46 | 65.90 | 83.22 | 68.11 | 75.00 | 74.15 | 50.27 | 64.79 |
| Qwen2VL-FT (ScoreRS-P30S30) | 77.03 | 70.49 | 79.92 | 70.92 | 89.50 | 77.57 | 82.50 | 69.20 | 84.52 | 75.05 | 80.00 | 78.25 | 54.22 | 66.24 |
| Qwen2VL-FT (CLIP-P30S60) | 80.17 | 71.69 | 82.49 | 70.68 | 92.40 | 79.49 | 82.60 | 66.90 | 85.71 | 92.67 | 82.00 | 81.98 | 53.01 | 66.03 |
| Qwen2VL-FT (ScoreRS-P30S60) | 85.66 | 74.86 | 89.49 | 73.33 | 92.80 | 83.23 | 82.00 | 68.90 | 89.68 | 86.63 | 87.00 | 82.84 | 55.58 | 66.58 |

Table 4: Comparison between our finetuned model (Qwen2VL-7B-RS) and existing VLMs. Our model is trained on quality-filtered data comprising the top 30% of pretraining samples and 60% of instruction samples from the VHM datasets, selected using ScoreRS as the quality assessment model

| | RS Classification | | | | | | RSVQA | | | | | | Grounding | General Knowledge |
|---|---|---|---|---|---|---|---|---|---|---|---|---|---|---|
| | AID | METERML | NWPU | SIRI-WHU | WHU-RS19 | Avg. | HR-Comp. | HR-Pres. | LR-Comp. | LR-Pres. | LR-R-U | Avg. | VG-DIOR | LHRS-Bench |
| LLaVA-1.6-7B | 52.83 | 44.78 | 44.70 | 59.08 | 69.30 | 54.14 | 68.60 | 64.40 | 64.32 | 56.84 | 61.00 | 63.03 | 41.59 | 64.78 |
| InternVL-2.5-8B | 64.50 | 57.17 | 59.17 | 57.66 | 80.90 | 63.88 | 75.50 | 65.80 | 71.16 | 66.21 | 72.00 | 70.13 | 15.39 | 65.86 |
| Qwen2VL-7B | 66.13 | 63.54 | 62.35 | 70.79 | 87.40 | 70.04 | 75.60 | 63.30 | 75.47 | 62.00 | 73.00 | 69.87 | 11.87 | 64.78 |
| LHRS-Bot-Nova | 83.06 | 72.74 | 83.97 | 72.21 | 96.20 | 81.64 | 89.30 | 87.60 | 88.11 | 83.89 | 79.00 | 85.58 | 31.51 | 52.46 |
| GeoChat | 73.47 | 34.87 | 89.37 | 53.04 | 85.30 | 67.21 | 83.30 | 59.10 | 90.52 | 90.63 | 97.00 | 84.11 | 19.77 | 36.23 |
| VHM | 92.03 | 74.33 | 94.76 | 70.62 | 96.50 | 85.65 | 83.30 | 68.30 | 90.11 | 89.89 | 87.00 | 83.72 | 55.99 | 33.04 |
| SkysenseGPT | 88.16 | 40.00 | 90.06 | 68.38 | 95.50 | 76.42 | 84.20 | 70.50 | 92.11 | 90.32 | 95.00 | 86.43 | 12.87 | 36.37 |
| Qwen2VL-7B-RS | 85.90 | 74.42 | 91.59 | 74.75 | 96.30 | 84.59 | 87.30 | 75.80 | 91.36 | 89.79 | 88.00 | 86.45 | 58.34 | 67.08 |

slightly worse on classification than using the complete dataset. However, when adopting a more moderate filtering approach (e.g., 60%), we observe substantial improvements: 3% on classification tasks, 5% on vision question answering, and 0.36% on challenging grounding and LHRS-Bench benchmarks. Notably, ScoreRS consistently outperforms CLIP score-based selection across all evaluation scenarios.

Based on these empirical findings, we strategically select the highest-scoring 30% of pretraining data and 60% of vision instruction data as ranked by ScoreRS, while scaling the LoRA rank size to 128. We then evaluate our finetuned model against leading RS-specific VLMs and state-of-the-art general-purpose VLMs. As shown in Table 4, our model, trained on high-quality data selected by ScoreRS, not only achieves comparable or superior performance on classification, visual question answering, and visual grounding tasks compared to RS-specific VLMs, but also outperforms general-purpose models on the LHRS-Bench benchmark. These results further confirm that data quality in RS vision-language pairs represents the primary bottleneck limiting the full potential of VLMs for RS image understanding. More discussion of these results can be found in the Appendix F.4.

## 4.2 Reinforcement Learning

We explore using ScoreRS as a reward model for RL training of our Qwen2VL-7B-RS model, using 8K samples (4K open-ended, 4K close-ended) from the filtered VHM vision instruction dataset. Inspired by Deepseek-R1 [19], we implement a two-step reasoning process with special tokens (<think>, </think> for reasoning; <answer>, </answer> for responses). Our reward functions include binary rewards for close-ended questions, ScoreRS-based rewards for open-ended questions (Section 3.3), and format rewards for proper structuring. We evaluate on vision grounding and LHRS-Bench tasks, which better assess RS image understanding capabilities than simpler tasks, comparing against non-RL-trained Qwen2VL-7B-RS and an RL version without ScoreRS rewards (where we substitute 4K additional close-ended questions to maintain equal training steps). Detailed settings are presented in Appendix D.5.

Table 5 shows that direct RL training on Qwen2VL-7B-RS (Qwen2VL-7B-RS-Zero) improves performance by 0.2% on LHRS-Bench and 1% on VG-DIOR. Notably, using ScoreRS for reward calculation outperforms the variant without ScoreRS-based rewards, confirming ScoreRS's effectiveness as a reward model. We also provide a comparison of this training paradigm with DPO and compare it with using GPT-4o as a reward model for open-ended questions in Appendix C.4.

Table 5: Comparison of RL-trained models. Qwen2VL-7B-RS-Zero: Directly apply RL to Qwen2VL-7B-RS. Qwen2VL-7B-RS-SFT: Qwen2VL-7B-RS fine-tuned with our manually generated reasoning data. Qwen2VL-7B-RS-R1: RL applied to Qwen2VL-7B-RS-SFT. "w/o ScoreRS": variant trained without ScoreRS-based rewards

| | VG-DIOR | LHRS-Bench |
|---|---|---|
| Qwen2VL-7B-RS | 58.34 | 67.08 |
| Qwen2VL-7B-RS-Zero (w/o ScoreRS) | 58.66 | 66.05 |
| Qwen2VL-7B-RS-Zero | 59.64 | 67.21 |
| Qwen2VL-7B-RS-SFT | 59.21 | 66.34 |
| Qwen2VL-7B-RS-R1 (w/o ScoreRS) | 62.47 | 65.71 |
| Qwen2VL-7B-RS-R1 | 64.52 | 69.13 |

During experiments, we observe that Qwen2VL-7B-RS-Zero exhibits overly simplistic reasoning patterns. To maximize RL training benefits, we implement a multi-stage approach: first, we used Qwen2VL-7B-RS-Zero to answer our RS-specific questions from Section 3.2.2, then manually refined these responses to create 2K curated reasoning-answer pairs. We fine-tune Qwen2VL-7B-RS on this dataset before applying RL training with the same data used for Qwen2VL-7B-Zero. The resulting model, Qwen2VL-7B-RS-R1, shows significant improvements—gaining over 5% on VG-DIOR and 2% on LHRS-Bench—while also outperforming variants trained without ScoreRS-based rewards. Analysis confirms more reasonable reasoning patterns, with conversation examples in Appendix 13.

### 4.3 Best-of-N Selection

We compare our ScoreRS and CLIP-score with LongCLIP [75] as a BoN selector Specifically, for each question, we generate multiple candidate answers and select the highest-scoring answer with given selector. We utilize three models for answer generation: the RS-specific LHRS-Bot-Nova, the general-purpose Qwen2VL-7B, and our fine-tuned Qwen2VL-7B-RS-R1. We evaluate on both VG-DIOR and LHRS-Bench datasets. Due to computational constraints and the large volume of the VG-DIOR dataset, we sample the first 1,000 instances for evaluation to reduce experimental costs. For all the answer generation, we set the sampling temperature and top-p parameters to 0.95 and 1, respectively.

Figure 4 demonstrates that using ScoreRS as a selector consistently enhances evaluation performance across models for both complex perception (VG-DIOR) and holistic vision understanding (LHRS-Bench) tasks, while CLIP-score underperforms in these scenarios. Notably, with ScoreRS as the BoN selector, accuracy on the challenging LHRS-Bench dataset exceeds 70% for the first time, highlighting

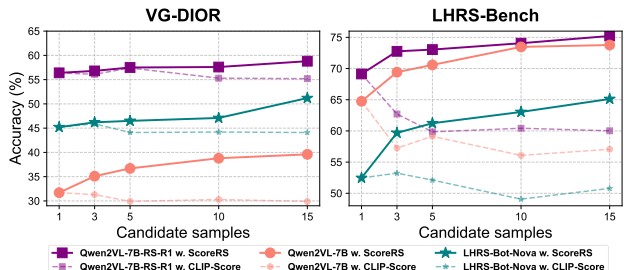

Figure 4: Comparison with different BoN selectors.

ScoreRS's potential not only for data selection but also for test-time scaling of base models in RS image understanding. We provide additional model validations and comparisons with majority voting in Appendix C.5.

### 4.4 Ablation Analysis

We validate our multi-stage training strategy and preference dataset quality through ablation studies on a held-out set of 6K samples (2K from image-caption preferences, 4K from vision instruction preferences). Performance is measured by accuracy, defined as the scoring model correctly assigning higher scores to preferred texts ($T^+$) over their counterparts ($T^-$). Additional ablation analyses examining score model size and the benefits of RS-specific VLM initialization are provided in the Appendix C.2.

We validate our preference dataset's effectiveness through joint training experiments comprising two stages: (1) initializing the value head with pure text preferences for stability, and (2) unfreezing both ViT and LLM components. As shown in Table 6, training with our complete preference dataset achieves the highest reward accuracy. Notably, omitting any subset degrades performance even when evaluating on

Table 6: Ablation study on multi-stage training strategy and preference dataset composition. We evaluate each dataset component's contribution (ICPD = Image-Caption Preference Dataset, VIPD = Vision Instruction Preference Dataset) and demonstrate our multi-stage approach's advantages over joint training alternatives.

| | Accuracy @ ICPD | Accuracy @ VIPD | Accuracy |
|---|---|---|---|
| Jointly Training | 83.13 | 81.07 | 82.09 |
| ✗ Image-Caption P.D. | 59.44 | 76.30 | 67.87 |
| ✗ Vision Instruction P.D. | 60.17 | 51.85 | 56.03 |
| Multi-Stage Training | **92.91** | **93.03** | **92.97** |

different domain (e.g., excluding image-caption preferences reduces accuracy on the vision instruction evaluation subset). Our multi-stage training strategy further improves reward accuracy by over 10% compared to joint training. These results confirm both the quality of our curated preference dataset and the benefits of our training approach.

# 5 Conclusion

Vision-language data serves as the fundamental building block for training VLMs. However, the quality control and data curation for RS-specific VLMs have not been fully addressed. In this study, we explore the development of parameter-wise scoring models for high-quality data selection. Through careful construction of RS preference datasets and the training of our ScoreRS model, we demonstrate that current vision-language datasets are far from optimal. Our findings show that using just 30% of quality-filtered data achieves superior performance compared to the complete dataset in both CLIP training and large VLM finetuning. Furthermore, we investigate the application of ScoreRS in RL training and BoN selections. Both applications demonstrate that ScoreRS can enhance VLM capabilities in solving complex and challenging tasks. We anticipate that this study will encourage the RS community to place more emphasis on vision-language data quality control and the strategic utilization of high-quality data.

# 6 Acknowledgement

This study was supported by the National Natural Science Foundation of China (Grant No. 42522112), the Natural Science Foundation of Jiangsu Province (Grant No. BK20250065), and the AI and AI for Science Project of Nanjing University (Grant No. 020914380171). We also would like to thank Haoqin Tu, Zhuo Zheng, and Mengke Zhu for their valuable discussions and feedback.

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

# Appendices

# A  Quality Dimension and Scoring System

## A.1  Quality Dimension

The detailed descriptions and clarifications for each dimension are provided below:

**Relevance:** The text accurately describes the content visible in the remote sensing image. It does not introduce objects, features, or relationships that are not present (i.e., avoids hallucination). All key elements mentioned in the text are verifiable in the image.

**Specificity & Detail:** The text provides specific and detailed information rather than being vague or overly general. It names identifiable objects, describes their characteristics (e.g., size, shape, color if discernible and relevant), and their spatial relationships.

**Completeness:** The text covers the most salient and important features of the image relevant to common RS tasks (e.g., land use classification, object detection, change detection). It doesn't omit obvious, critical elements. For question-answering, the answer should address the question if the information is present.

**Clarify & Fluency:** The text is grammatically correct, well-structured, and easily understandable. It uses clear, complete, and unambiguous language.

**Semantic Richness:** The text captures semantic information that is particularly useful for downstream RS applications. This could include types of land cover (e.g., "deciduous forest," "industrial zone," "low-density residential"), specific object classes ("solar farm," "roundabout," "cooling towers"), or activities ("active construction site").

**Distinction Between Specificity & Detail, Completeness, and Semantic Richness:**  We delineated three potentially overlapping quality dimensions to ensure non-redundant evaluation:

(1) *Specificity & detail* focuses on the level of precise information provided (e.g., "high-rise commercial buildings" vs. just "buildings"). It's about granularity and precision in what is described.

(2) *Completeness* focuses on coverage of all important elements in the image. It's about breadth - whether all significant features are mentioned, regardless of how detailed each mention is.

(3) *Semantic Richness* focuses on the use of domain-specific terminology and concepts relevant to remote sensing (e.g., "center-pivot irrigation" vs. "circular fields").

These dimensions can vary independently—a description might be highly specific yet incomplete, complete yet lacking domain terminology, or semantically rich yet insufficiently detailed—justifying their separate evaluation.

## A.2  Scoring System

We provide our detailed scoring criteria for image-caption pairs in Table 16 and for instruction samples in Tables 17 and 18.

# B  Additional Detail for Preference Data Generation

## B.1  Deduplication

**Image-Caption:**  We implement a rigorous image deduplication process for the LHRS-Align [39] to ensure image representativeness for building our image-caption preference dataset: (1) Feature extraction using the SSCD copy detection model [45, 18] to compute image embeddings. (2) Similarity computation through cosine distance computation between image embeddings, with a carefully tuned threshold of 0.65 (empirically determined through experiments across the range 0.6-0.9). (3) Duplicate grouping via connected-components algorithm, preserving one image-text pair per component. Finally, this deduplication pipeline yields 76K distinctive representative RS images.

**Vision Instruction:**  We employ a two-stage similarity-based filtering process to ensure diversity of instruction sample: (1) Text similarity: let $Q = \{T_1, T_2, \ldots, T_n\}$ be the set of all questions extracted from the conversations. We compute their semantic embeddings using the BGE [6] model, where for

any two queries $T_i, T_j \in Q : \text{sim}_{\text{text}}(T_i, T_j) = \cos(\text{BGE}(T_i), \text{BGE}(T_j))$; (2) Image similarity: For query pairs where $\text{sim}_{\text{text}}(T_i, T_j) > 0.65$, we further examine their corresponding images $I_i, I_j$ using SSCD [45] embeddings: $\text{sim}_{\text{image}}(I_i, I_j) = \cos(\text{SSCD}(I_i), \text{SSCD}(I_j))$. When both $\text{sim}_{\text{text}}(T_i, T_j)$ and $\text{sim}_{\text{image}}(I_i, I_j)$ exceed 0.65, we retain only one question-image sample from the pair to minimize redundancy.

## B.2  Model Selection

For the preference generation policy models, we aimed to maximize diversity. However, considering the high costs associated with adding multiple models (including generation and evaluation costs), we selected 3 policy models for image-caption preference dataset generation and 2 for vision instruction preference dataset creation.

Regarding model families, we ensured inclusion of two model types: RS-specific VLMs and general VLMs, to increase answer diversity. For example, RS-specific VLMs typically include domain-specific terminology such as "*high resolution, true color*", while general VLMs often add contextual descriptors like "*aerial, satellite*".

For RS-specific VLMs, we selected LHRS-Bot-Nova [30] based on its superior performance in our preliminary evaluations. For general VLMs, we aim to select the highest-performing available models. At the time of dataset preparation, the best open-source VLMs are Qwen2VL and InternVL2.5, so we selected these model families using the same parameter scale (7B) as LHRS-Bot-Nova for policy generation.

We acknowledge that model size, model family, prompting strategies, and generation hyperparameters all influence the final preference dataset [29, 23], and synthetic data generation itself represents a critical research topic [41, 2]. In this work, we focused primarily on a reasonable model selection strategy while emphasizing diversity and domain specificity of the RS images and instructions used to generate preference data. We leave for future work the exploration of how different policy models affect preference data quality and scoring model performance.

## B.3  Generation Parameters and Prompts

### B.3.1  Image-Caption Preference Dataset

**Caption Generation Setting**  We employ three state-of-the-art VLMs—LHRS-Bot-Nova, Qwen2VL-7B, and InternVL-2.5-8B—to generate captions for the given images. For all models, we use the prompt: *Provide a factual description highlighting important details in this picture.* The generation temperature parameter for LHRS-Bot-Nova was set to 0.7, while for Qwen2VL-7B and InternVL-2.5-8B, we maintain their standard generation configuration parameters.

**Caption Judgment Setting**  For the selection of captions in our preference pairwise data, we employ GPT-4o (GPT-4o-2024-05-13). The complete prompt used for this judgment is provided in Table 19.

**Data Samples**  We provide several representative samples from our image-caption preference dataset in Figure 10 to illustrate the quality of the collected data.

### B.3.2  Vision Instruction Preference Dataset

**Judgment Setting**  We provide the prompt in Table 20 for using Qwen2VL-72B as judger to select the better answer for the input question.

**RS Specific Category and Question**  We manually design RS specific questions across 12 categories: *basic visual recognition, spatial analysis, environmental assessment, urban analysis, agricultural analysis, disaster assessment, geological features, infrastructure analysis, temporal understanding, advanced reasoning, quantitative assessment, and color spectral analysis*. These domain-specific questions are presented in Table 21 and Table 22.

**Generation Prompt**  We utilized LHRS-Bot-Nova and Qwen2VL-7B to answer the RS-specific questions. The prompt template employed for this question-answering task is presented in Table 23.

Table 7: Human validation of preference judgments. 'Model-Human Agreement' measures the model's accuracy against human judges. 'Inter-Annotator Agreement' measures the consistency among human judges on a shared subset of samples.

| Model-Human Agreement | | Inter-Annotator Agreement | |
|---|---|---|---|
| Accuracy @ ICPD | Accuracy @ VIPD | Human Consistency @ ICPD | Human Consistency @ VIPD |
| 90.06% (1351/1500) | 96.80% (1452/1500) | 88% (44/50) | 96% (48/50) |

**Rephrase Prompt**  We employ Qwen2.5-13B to rephrase the manually designed questions to enhance question diversity. The prompt used for this rephrasing process is presented in Table 24.

**Data Samples**  We provide serveal samples from our vision instruction preference dataset in Figure 11.

## B.4  Further Human Validation of Preference Data

While powerful, using general-purpose models like GPT-4o and Qwen2VL-72B for preference judgments warrants a careful validation of their accuracy. We chose these models over existing domain-specific VLMs (which are typically smaller, 13B parameters) because they demonstrate superior instruction-following capabilities for the nuanced task of judging responses. For instance, our preliminary human validation showed that GPT-4o achieved 92.6% agreement, significantly outperforming a domain-specific model like LHRS-Bot-Nova (82.8%). Despite limited direct exposure to remote sensing data, the diverse training of GPT-4o and Qwen2VL-72B provides the robust, context-dependent reasoning required for this task.

To further validate the accuracy of our preference dataset, we conducted a dedicated human annotation experiment. We selected 3,000 samples of preference data, distinct from those in our main manuscript validation, comprising 1,500 from the image-caption preference dataset (ICPD) and 1,500 from the vision-instruction preference dataset (VIPD). Three human annotators were tasked with a binary judgment (Yes/No) on whether the chosen positive sample was qualitatively better than the negative sample. Additionally, to assess the reliability of our human evaluation, all three annotators evaluated a shared set of 100 common samples (50 from ICPD, 50 from VIPD) to establish inter-annotator agreement.

The results, summarized in Table 7, show that the judgments from our models align well with human preferences. The higher consistency and model alignment on VIPD tasks suggest that instruction-based responses are easier to evaluate than the more subjective, lengthy image captions, where human preferences can naturally vary. While this curation strategy proves effective, we acknowledge it is not perfect. Future work could incorporate more complex methods, such as Retrieval-Augmented Generation (RAG) with human-annotated data or self-reflection mechanisms, to further refine judgment quality.

## C  Additional Result

### C.1  Score Model Training

Directly using the output from the final value head as a score creates challenges in defining the range, which affects not only the relative importance of data but also reward calculation in RL training (notably, algorithms like GRPO typically require a reward range of 0-1). To address this issue, we attempt to resolve it at the source by adding normalizing functions after the value head output.

We experiment with sigmoid, tanh, and softplus functions (though softplus only ensures positive scores). The training loss is presented in Figure 5. Despite trying different initializations for the final value head, we observe that using tanh and sigmoid as normalization functions led to easy convergence during training (which we interpret as the scores lying in the saturated region of these functions). Furthermore, since we observe that softplus performed worse than using no normalization at all (labeled as "raw" in Figure 5), we opt to use the latter setting during the score model training.

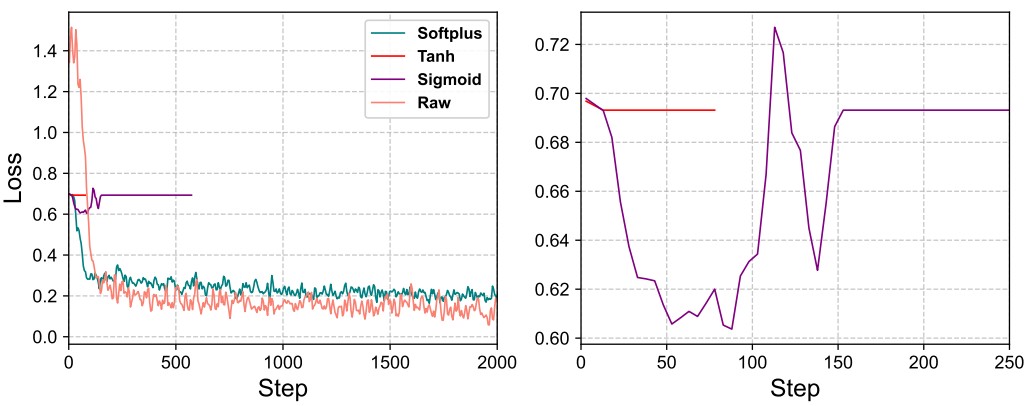

Figure 5: Training loss of score model with different normalization functions

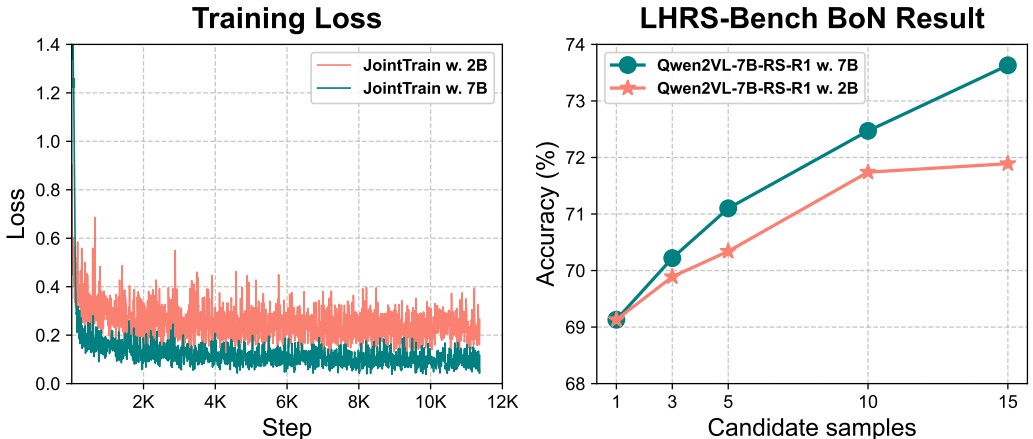

Figure 6: Comparison of scoring models with different sizes on training loss and performance as BoN selector for LHRS-Bench

Table 8: Ablation analysis on the effectiveness of scoring model size for distinguishing between good and bad text sample on the hold-out preference set. ICPD = Image-Caption Preference Dataset, VIPD = Vision Instruction Preference Dataset

|  | Accuracy @ ICPD | Accuracy @ VIPD | Accuracy |
|---|---|---|---|
| Jointly Training w. 7B | **83.13** | **81.07** | **82.09** |
| Jointly Training w. 2B | 71.53 | 79.63 | 75.58 |

## C.2 Ablation Study

### C.2.1 Influence of Model Size

We analyze the impact of initialized model size on the effectiveness of our scoring model. Considering the computational cost for full-parameter tuning of models larger than 7B, we evaluate the smaller Qwen2VL-2B model as a comparison to Qwen2VL-7B. For simplicity, we compare the results using the joint training strategy introduced in Section 4.4. To ensure a fair comparison, we use the same model family (Qwen2VL) and identical optimization steps (i.e., same batch size). We sweep through a wide range of learning rates from $1 \times 10^{-7}$ to $1 \times 10^{-4}$ for Qwen2VL-2B, and find that $8 \times 10^{-5}$ yields the best results (according to the training loss), compared to $1 \times 10^{-6}$ for Qwen2VL-7B.

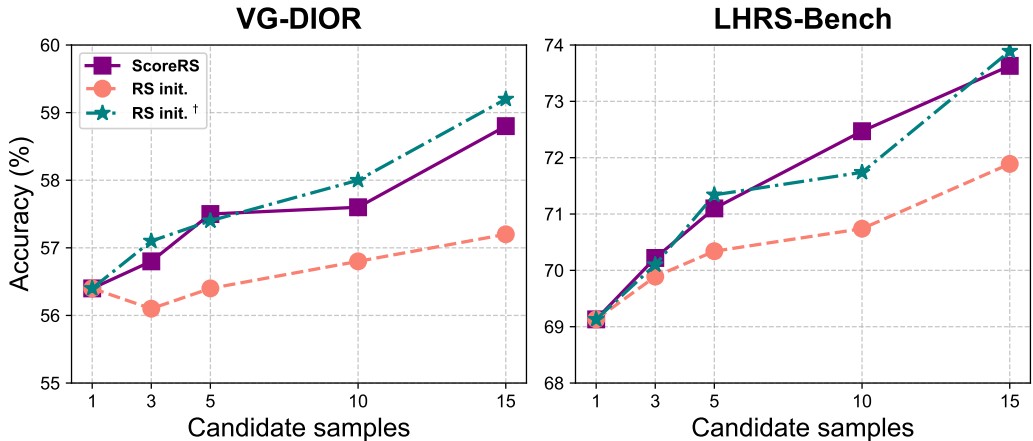

Figure 7: BoN comparison on different score model with different initialzietion base model. ScoreRS is initialized with Qwen2VL-7B, RS init. is initialized with Qwen2VL-FT from Table 3, and RS init.[†] is initialized with Qwen2VL-FT (ScoreRS-P30S60) from Table 3

Table 9: Comparison of different initialization strategies for the scoring model. We initialize scoring models with different base models, use them to filter the VHM instruction dataset with a 60% selection ratio, and fine-tune Qwen2VL-FT (ScoreRS-P30) from Table 3 with the filtered data. ScoreRS is initialized with Qwen2VL-7B, RS init. is initialized with Qwen2VL-FT from Table 3, and RS init.[†] is initialized with Qwen2VL-FT (ScoreRS-P30S60) from Table 3

| | RS Classification | | | | | | RSVQA | | | | | | Grounding | General Knowledge |
| | AID | METERML | NWPU | SIRI-WHU | WHU-RS19 | Avg. | HR-Comp. | HR-Pres. | LR-Comp. | LR-Pres. | LR-R-U | Avg. | VG-DIOR | LHRS-Bench |
|---|---|---|---|---|---|---|---|---|---|---|---|---|---|---|
| Qwen2-VL-FT (ScoreRS) | **85.66** | 74.86 | **89.49** | 73.33 | 92.80 | 83.23 | 82.00 | **68.90** | **89.68** | 86.63 | **87.00** | 82.84 | 55.58 | 66.58 |
| Qwen2-VL-FT (RS init.) | 82.43 | 73.52 | 88.60 | **74.56** | 92.40 | 82.30 | 83.20 | 67.20 | 86.33 | **91.40** | 82.00 | 82.03 | 53.47 | 66.29 |
| Qwen2-VL-FT (RS init.[†]) | 84.90 | 75.38 | 89.20 | 74.55 | 94.60 | **83.73** | **83.40** | **68.90** | 88.87 | 90.96 | 86.00 | **83.63** | **56.21** | **66.90** |

We compare the two variants' training loss, accuracy on the held-out preference set as described in Section 4.4, and their performance as BoN selectors. From the training loss and BoN results shown in Figure 6, we observe that the smaller model converges at a higher loss compared to the larger model. On LHRS-Bench as a selector, the smaller selector performs inferior to the larger selector and tends to saturate when given more candidate samples, indicating that smaller models struggle to identify the best candidate among different samples. On the hold-out preference evaluation set presented in Table 8, we see that the 2B model performs approximately 7% worse compared to the 7B model, further demonstrating its reduced effectiveness in distinguishing good responses from bad ones compared to the larger model.

These results confirm that larger scoring models indeed provide better performance, which aligns with related work [17]. We anticipate that future work could develop even larger, more advanced scoring models to bring additional benefits to the RS community.

### C.2.2 Influence of Model Initialization

Next, we tackle a critical question: whether initializing the scoring model with an RS-specific VLM instead of a general VLM (QwenVL2-7B in our case) provides benefits. We use our fine-tuned Qwen2VL-FT from Table 3 (i.e., the model fine-tuned with all VHM datasets) to initialize the scoring model and conduct the three-stage training with the same preference data. In this setting, we maintain the original base architecture (Qwen2VL) and training recipe to ensure a fair comparison. For evaluation, we use the different scoring models to filter the VHM vision instruction dataset with a fixed selection ratio of 60%, and use the filtered instruction dataset to fine-tune Qwen2VL-FT (ScoreRS-P30) from Table 3. The fine-tune setting is the same as Section D.4. We also evaluate them as BoN selectors on VG-DIOR and LHRS-Bench with our Qwen2VL-7B-RS-R1. The BoN evalution setting is the same as Section 4.4.

The results are presented in Table 9 and Figure 7. From these results, we observe that initializing the scoring model with an RS-specific VLM yields inferior performance in both data filtering quality and BoN selection compared to using the general Qwen2VL as the base model. We hypothesize

Table 10: Comparison of different finetuned CLIP models on classification tasks

| | NWPU@1 | NWPU@5 | EuroSAT@1 | EuroSAT@5 | fMoW@1 | fMoW@5 | AID@1 | AID@5 | SIRI-WHU@1 | SIRI-WHU@5 | WHU-RS19@1 | WHU-RS@5 | Avg.@1 | Avg.@5 |
|---|---|---|---|---|---|---|---|---|---|---|---|---|---|---|
| CLIP | **65.31** | **93.23** | 42.14 | 89.20 | **29.40** | **60.21** | 64.11 | 91.21 | **58.11** | 85.17 | **86.24** | **99.21** | 57.55 | 86.37 |
| Skyscript (ALL) | 48.11 | 76.36 | 50.60 | 85.13 | 19.07 | 45.76 | 51.81 | 77.44 | 43.29 | 82.29 | 62.49 | 93.83 | 45.90 | 76.80 |
| Skyscript (30% w. CLIP-Score) | 60.53 | 90.34 | 58.60 | 93.44 | 26.36 | 52.28 | 59.67 | 84.40 | 47.75 | 84.50 | 80.60 | 98.01 | 55.59 | 83.83 |
| Skyscript (30% w. ScoreRS) | 63.43 | 92.54 | 60.79 | **96.99** | 29.32 | 59.69 | **64.59** | **91.56** | 55.02 | **85.91** | 84.27 | 98.96 | **59.57** | **87.61** |

Table 11: Performance of different data filtering methods at various saving ratios. Lower saving ratios indicate more extreme data filtering.

| Saving Ratio | Method | Classification | Retrieval |
|---|---|---|---|
| 30% | ScoreRS | 70.97 | 27.11 |
| | CLIP-Score | 69.98 | 26.36 |
| 20% | ScoreRS | 68.44 | 27.01 |
| | CLIP-Score | 67.70 | 26.69 |
| 10% | ScoreRS | 63.19 | 26.16 |
| | CLIP-Score | 62.89 | 26.05 |

that this occurs because Qwen2VL-FT is fine-tuned with the complete VHM dataset, which likely contains low-quality vision-language data that biases the base model toward a domain that is difficult to optimize for high-quality sample selection.

To test this hypothesis, we initialize the scoring model with an RS-specific VLM that is fine-tuned with ScoreRS-filtered data—specifically, Qwen2VL-FT (ScoreRS-P30S60) from Table 3. With this modification, the new scoring model (RS init.[†]) performs better on both data filtering and as a BoN selector, which supports our hypothesis.

Our analysis indicates that if we can ensure the RS-specific VLM is of high quality (though "high quality" may be difficult to precisely define, it can be approximated through data that has undergone cleaning through a qualified pipeline), specialized initialization of the scoring model will yield better performance.

## C.3 CLIP Training

### C.3.1 Evaluation on Skyscript Dataset

We extend our investigation by applying ScoreRS to select data from the larger-scale RS image-caption dataset Skyscript [62]. Except for the dataset used, all experimental details remain identical to our RemoteCLIP fine-tuning setup.

The results, presented in Table 10, further demonstrate that filtering with our ScoreRS yields superior performance compared to both using the complete dataset and applying CLIP-score filtering. We observe that directly fine-tuning CLIP with the entire Skyscript dataset actually resulted in performance inferior to the original CLIP model. We attribute this to the rule-based caption construction method used in Skyscript, which introduces significant redundancy and thus necessitates more careful data selection for effective fine-tuning.

### C.3.2 Evaluation on Extreme Data Filtering Scenarios

We investigate the impact of extreme data filtering by evaluating performance at various data saving ratios, as shown in Table 11. The results demonstrate that data quantity remains crucial for maintaining high performance, particularly for classification tasks. Extreme filtering ratios significantly degrade classification accuracy, likely because datasets with numerous classes require a sufficient number of samples per class to learn effectively. In contrast, retrieval tasks appear more resilient to data reduction. This suggests that even a small, high-quality subset of data can effectively capture the essential text-to-image alignment concepts required for retrieval, thus maintaining comparable performance even under extreme pruning.

Table 12: Performance comparison of different filtering methods on the curated RemoteCLIP dataset (30% saving ratio). The classification benchmark includes NWPU, AID, METERML, WHU-RS19, SIRI-WHU, fMoW, and EuroSAT. The retrieval benchmark uses UCM and RSICD.

| Method | Classification Avg @ 1 | Retrieval Avg @ 1 |
|---|---|---|
| ScoreRS | **70.97** | **27.11** |
| CLIP-Score | 69.98 | 26.36 |
| SigCLIP-2 | 68.24 | 26.47 |
| CLIP-LAION-RS | 70.59 | 26.41 |

Table 13: Comparison between ScoreRS-based GPRO and DPO RL training. DPO/8K represents training with the same data scale as GPRO training, while DPO/26K represents training with the complete preference dataset for DPO training

| | VG-DIOR | LHRS-Bench |
|---|---|---|
| Qwen2VL-7B-RS-DPO/8K | 58.97 | 66.45 |
| Qwen2VL-7B-RS-DPO/26K | **59.71** | 67.19 |
| Qwen2VL-7B-RS-Zero | 59.64 | **67.21** |

### C.3.3 Comparison with State-of-the-Art Filtering Methods

To further evaluate the effectivness of our ScoreRS, we conduct an analysis using robust, publicly available models as baselines. We noted that CLIP-LAION-RS [62], which was fine-tuned on a remote sensing (RS) subset of the well-curated LAION-2B dataset, outperforms standard CLIP-filtered versions. We, therefore, curated the RemoteCLIP dataset using scores from both SigCLIP-2 (L/16-256) and CLIP-LAION-RS (L/14), comparing their filtering performance against our ScoreRS method. For this experiment, we used a fixed data saving ratio of 30%.

The results are presented in Table 12. Although SigCLIP-2 generally exhibits strong performance on various tasks compared to the original CLIP models, factors such as data distribution and training recipes may lead to different outcomes in the specialized remote sensing domain. The SigCLIP-2 filtered version shows inferior performance on classification tasks, while the CLIP-LAION-RS filtered version shows a marked improvement over the standard CLIP-Score. Notably, our ScoreRS filter demonstrates the best performance among all baselines, achieving the highest scores in both classification and retrieval.

From these experiments, it is clear that ScoreRS is the first scoring model in the RS community capable of effectively evaluating diverse types of vision-language data. Additionally, thanks to the strong capabilities of its Qwen2VL backbone, it can be extended to evaluate complex multi-image vision-language data, such as change captioning tasks.

### C.4 RL Comparison

### C.4.1 Compare with DPO

We compare the effectiveness of using ScoreRS as a reward function for open-ended questions with GPRO against Direct Preference Optimization (DPO)[47]. Specifically, we use the 26K vision instruction preference pairs (excluding the RS-specific preference data) constructed in Section 3.2.2 for DPO training. Starting with our fine-tuned Qwen2VL-7B-RS model (Table 5), we conduct full-parameter DPO training using LLaMA-Factory and compare the results with Qwen2VL-7B-Zero (Table 5). For a fair comparison, we train two DPO variants: one using 8K preference pairs (comprising 4K open-ended questions and 4K closed-ended questions), and another using the full 26K preference dataset. We evaluate the resulting models on VG-DIOR and LHRS-Bench evaluation datasets.

The results are presented in Table 13. We observe that with equivalent data scale, DPO performs inferior to GPRO training, while increasing the data scale for DPO yielded performance on par with or slightly better than GPRO training. This observation aligns with the established conclusion that RL training is typically more data-efficient than SFT [5, 11] (here, we consider DPO as a generalized form of SFT). In our case, the data efficiency advantage is approximately 3× (8K vs. 26K).

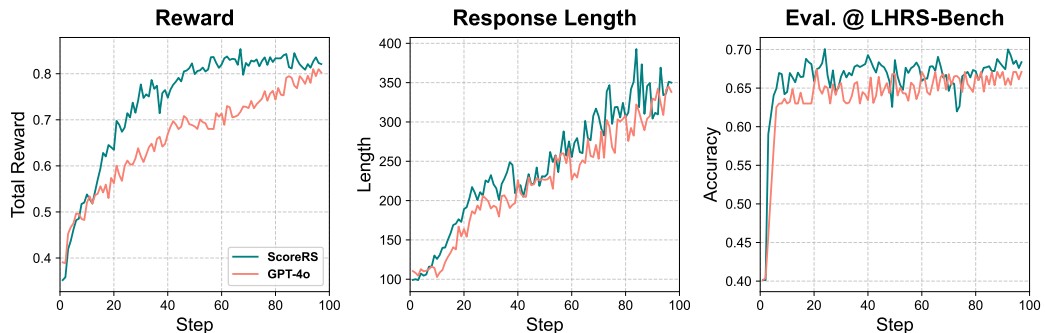

Figure 8: Comparison with GPT-4o as reward function for open-ended questions

Table 14: Comparison between Qwen2VL-7B-RS and its PPO training variant

| | RS Classification | | | | | RSVQA | | | | | Grounding | General Knowledge |
|---|---|---|---|---|---|---|---|---|---|---|---|---|
| | AID | METERML | NWPU | SIRI-WHU | WHU-RS19 | Avg. | HR-Comp. | HR-Pres. | LR-Comp. | LR-Pres. | LR-R-U | Avg. | VG-DIOR | LHRS-Bench |
| Qwen2VL-7B-RS | 85.90 | 74.42 | 91.59 | 74.75 | 96.30 | 84.59 | 87.30 | 75.80 | 91.36 | 89.79 | 88.00 | 86.45 | 58.34 | 67.08 |
| Qwen2VL-7B-RS-PPO | 84.76 | 69.03 | 90.68 | 70.79 | 92.60 | 81.57 | 85.70 | 72.80 | 86.32 | 86.85 | 92.00 | 84.73 | 57.10 | 66.04 |

### C.4.2 Compare with GPT-4o as Reward Model

Since our preference dataset is largely built using GPT-4o as a judge (excluding closed-ended questions) based on our scoring system, a natural question arises: would using the same scoring system with GPT-4o as the reward model for calculating rewards for open-ended questions be more effective? To investigate this, we utilize GPT-4o for reward calculation based on our scoring system and normalized the total reward (e.g., average reward / 5) as the final reward for open-ended questions. We start with the Qwen2VL-7B-RS model from Table 5 and use the same training dataset and experimental settings as described in Section 4.2. We compare these results with those obtained using our ScoreRS(i.e., compared with Qwen2VL-7B-RS-Zero).

We plot the total reward, response length, and evaluation results on LHRS-Bench during the training process in Figure 8. The results show that despite our dataset being largely built with GPT-4o, using GPT-4o as the open-ended reward model performed inferior to using our ScoreRS. We suspect two main reasons for this outcome. First, since our ScoreRS is fine-tuned with large-scale RS-specific preference data, although the dataset is built using GPT-4o judgments, it acquires substantial domain-specific knowledge (or simply put, developed a bias toward this specific domain) that helps it better distinguish between good and better responses. Second, the closed-ended preference dataset enables specific capabilities in ScoreRS. Moreover, we should note that using GPT-4o as a reward model during the training process is significantly more costly than using ScoreRS (here, cost refers to money, with the single training run costing approximately $360).

### C.5 BoN Selection

We further evaluate our ScoreRS on more advanced VLMs, including InternVL3-8B [55] and Qwen2.5VL-7B [56]. To validate the effectiveness of our approach, we compare it with majority voting (i.e., selecting the most common candidate as the final answer). The generation sampling parameters are the same as those described in Section 4.3. The results are presented in Figure 9.

Our findings clearly demonstrate that even with more advanced (i.e., more optimized) models, our ScoreRS can boost performance as a BoN selector and outperform the majority voting selection mechanism. Notably, for the RS vision grounding task, since the bounding boxes generated vary across different candidate samples, majority voting often fails to improve model performance, while our ScoreRS consistently delivers improvements on this task.

### C.6 PPO Training

We explored using our ScoreRS directly as a reward model for RL through proximal policy optimization (PPO) [50]. We employ our finetuned Qwen2VL-7B-RS as the policy model, while the critic model is initialized with the same Qwen2VL-7B-RS architecture but with its language head

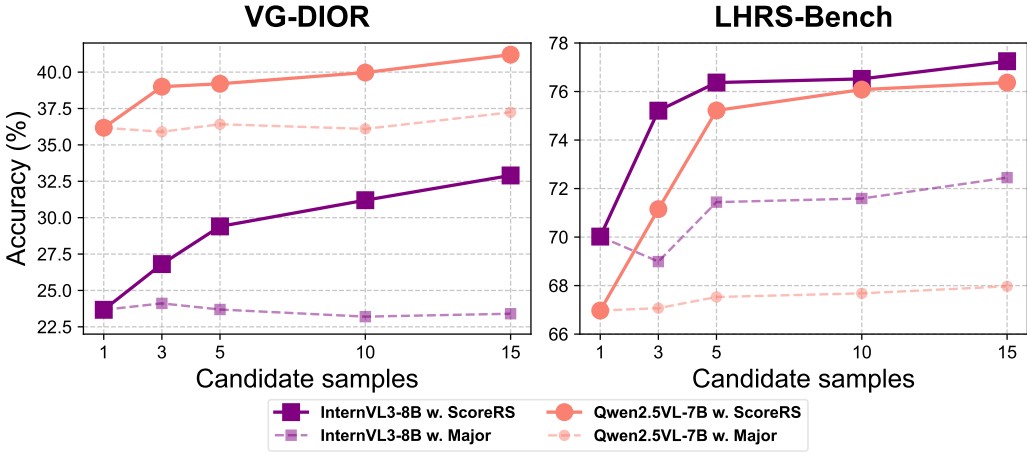

Figure 9: BoN comparison on more advanced models with ScoreRS as selector compared to majority voting

replaced by a learnable linear layer. We utilize the same dataset described in Section 4.2 for this training process. LoRA is applied to all linear layers in the policy model with rank and $\alpha$ parameters both set to 64. For text generation, we configure a sampling temperature of 0.95 and top-p of 0.9, with maximum new token generation limited to 512 tokens. The implementation uses a rollout batch size of 16, with PPO buffer size of 8 and 4 PPO epochs. We initialize the KL penalty coefficient at 2 and gradually increase it to 6 throughout the training process. The $\lambda$ parameter for Generalized Advantage Estimation (GAE) [49] is set to 0.95. For optimization, we use a learning rate of $1 \times 10^{-5}$ with a 0.1 warmup ratio and cosine learning rate decay schedule.

The results presented in Table 14 reveal that the model after PPO training performs inferiorly to its base variant across almost all evaluation datasets. We suspect this underperformance stems from suboptimal hyperparameter selection and reward calculation. Since we directly use our ScoreRS as reward model without any normalization or reference-based approach, this may be caused by reward hacking [1]. Moreover, given the considerable number of parameters associated with the PPO algorithm, we expect that developing comprehensive reward calculation strategies and conducting thorough hyperparameter searches remain important directions for future work.

# D    Experimental Setting

## D.1    Hardware and Framework

All our experiments are conducted on 2 nodes, each equipped with $8 \times$ A100-80G GPUs. For training the scoring model, we develop a customized training framework based on the OLMo training framework[3]. For fine-tuning the large VLMs, we utilized the LLaMA-Factory training framework[4]. For RL training, we customized our framework based on the VeRL training framework[5].

We implement our scoring model as an API call for reward calculation during the RL training process. During our RL training, we observe that the bottleneck is the reward calculation process when using a single scoring model to evaluate rollouts from different actors. To maximize training efficiency, we deploy multiple scoring models on different GPUs in a different node (specifically, 8 scoring models in our case), and each actor calls a specific scoring model according to its local rank. This approach significantly increased our RL training efficiency.

Table 15: Hyperparameter for training our ScoreRS

|  | Stage 1 | Stage 2 | Stage 3 |
|---|---|---|---|
| Batch Size | 64 | 16 | |
| Weight Decay | 0 | 0.1 | |
| Learning Rate | $2 \times 10^{-5}$ | $1 \times 10^{-6}$ | |
| WarmUp Iter | | 500 | |
| Epoch | | 1 | |
| Gradient Accumulation | 1 | 2 | 4 |

## D.2    ScoreRS Training

We implement a three-stage training approach for our ScoreRSmodel. Throughout all stages, we employ the AdamW optimizer with cosine learning rate decay and set $(\beta_1, \beta_2)$ to $(0.9, 0.95)$. For computational efficiency, we implement ZeRO-2 optimization with bfloat16 precision across all training stages. Additional hyperparameters are detailed in Table 15.

## D.3    CLIP Finetuning

We utilize CLIP-ViT-L/14 from Hugging Face[6] as our base model. Across all fine-tuning experiments, we employ ZeRO-2 optimization with bfloat16 precision and the AdamW optimizer.

For fine-tuning on RemoteCLIP, we configure a batch size of 1,024, learning rate of $1 \times 10^{-5}$, weight decay of 1.0, and warmup iterations of 200. The model is fine-tuned for 5 epochs using cosine learning rate decay.

For fine-tuning on Skyscript, we use a larger batch size of 2,048, learning rate of $1 \times 10^{-5}$, weight decay of 0.01, and warmup iterations of 2,000. The model is fine-tuned for 20 epochs with a fixed learning rate schedule, reducing the rate by a factor of 0.316 at 80% and 90% of the training process.

### D.3.1    Evaluation Setting

We evaluate the fine-tuned CLIP model on both RS classification and RS image-text retrieval tasks. For the classification tasks, we employ text prompts in the format of "*a satellite photo of {class name}*" and "*a satellite image of {class name}*" to perform the classification.

**Evaluation Metric:**    For the classification task, we report top-1 and top-5 accuracies. For the retrieval task, we evaluate both image-to-text and text-to-image performance using top-1 and top-5 recall metrics.

**Evaluation Dataset:**    (1) **NWPU (NWPU-RESISC45)** [9] contains 31,500 RGB images devided into 45 scene classes, each class containing 700 images. We use the entire dataset for evaluation. (2) **EuroSAT** [20] is based on Sentinel-2 satellite imagtes consisting out of 10 classes with in total 27,000 labeled images. We use the entire dataset for evaluation. (3) **fMoW** [10] contains 1,047,691 images convering 207 country and the total images are devided into 63 categories. We use the test split of fMoW-RGB for evaluation (16,948 samples). (4) AID [67] contains 10,000 images from 30 different categories. We use the entire dataset for evaluation. (5) **SIRI-WHU** [37] is a classfication dataset that contains 2,400 images from 12 classes, with each class has 200 images. We use the entire dataset for evaluation. (6) **WHU-RS19** [13] contains 19 classes, with each class has about 50 samples. We use entire dataset for evaluation. (7) **UCM (UCM-Caption)** [34] contains 2,100 images, with each image has 5 different captions. We use the same retrieval evaluation recipe as Skyscript [62]. (8) **RSICD** [34] contains 10,921 images, with each image has 5 different captions. We use the same retrieval evaluation recipe as Skyscript [34].

---

[3]https://github.com/allenai/OLMo
[4]https://github.com/hiyouga/LLaMA-Factory
[5]https://github.com/volcengine/verl
[6]https://huggingface.co/openai/clip-vit-large-patch14

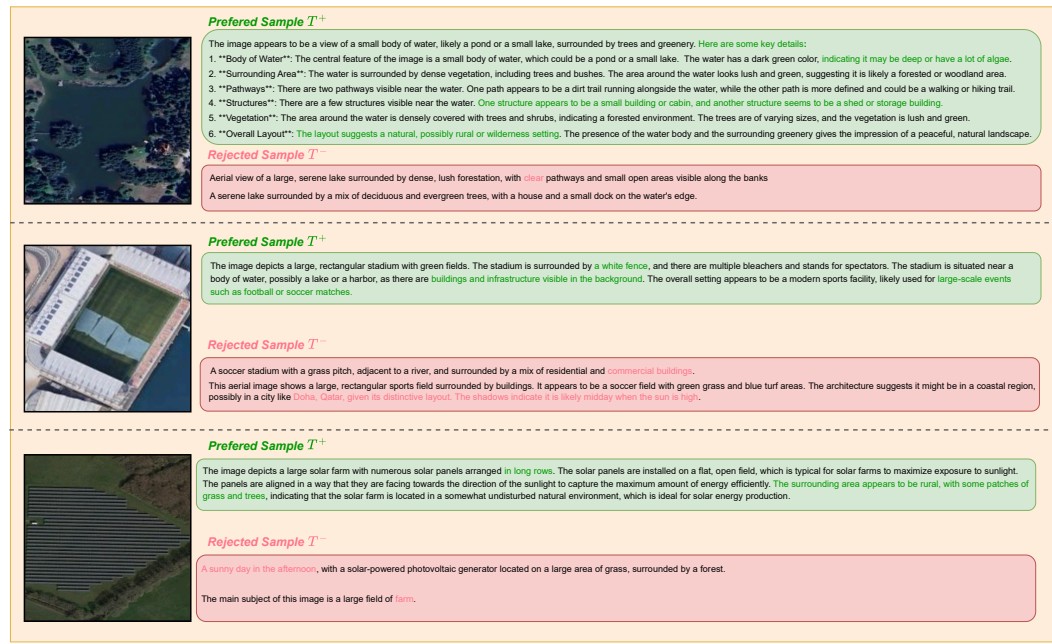

Figure 10: Representative examples from our image-caption preference dataset. Green represents reasonably good expression, while red represents low-quality expression

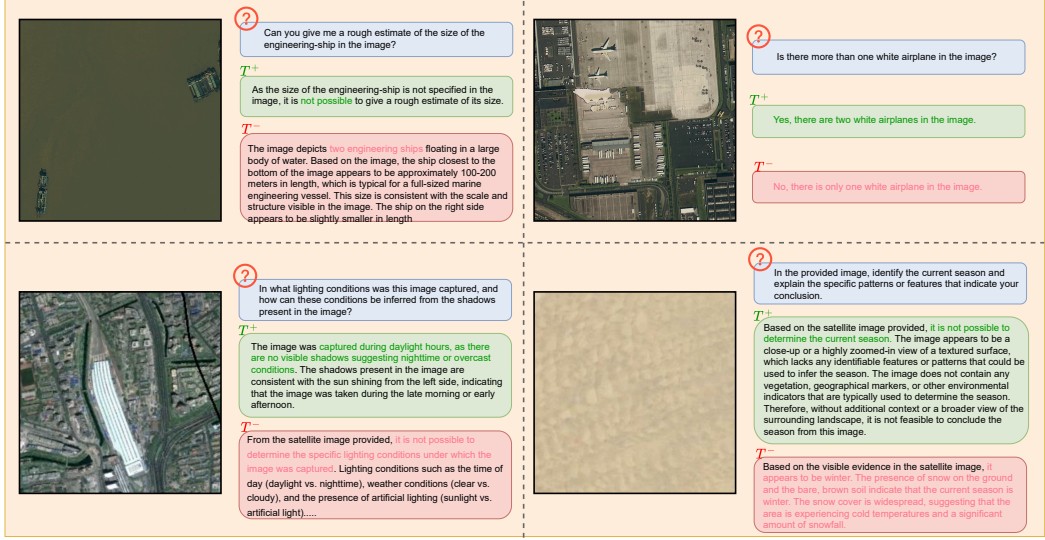

Figure 11: Representative examples from our vision instruction preference dataset. Green represents reasonably good expression, while red represents low-quality expression

## D.4 Large VLMs Finetuning

We select Qwen2VL-7B-Instruct as our base model and utilize the RS image-caption and vision instruction data from VHM as our finetuning dataset. Due to special token format differences between the VHM dataset and Qwen2VL-7B, we implement rule-based conversion methods to align the special tokens with Qwen2VL-7B requirements.

Our fine-tuning approach consists of two sequential stages: 1. In the first stage, we finetune the model using image-caption data while only unfreezing the vision-language connector. 2. In the second stage, we finetune using vision instruction data, applying LoRA adaptation to the base LLM while maintaining the unfrozen vision-language connector.

For both stages, we employ the AdamW optimizer with ZeRO-2 optimization strategy, bfloat16 precision, a maximum context length of 8,192, and cosine learning rate scheduling. Each stage is trained for 1 epoch.

Stage-specific hyperparameters are as follows:

- First stage: learning rate of $8 \times 10^{-5}$, batch size of 64, weight decay of 0.01, warmup ratio of 0.1, and maximum image resolution of 768.

- Second stage: learning rate of $2 \times 10^{-4}$, batch size of 64, weight decay of 0.01, warmup ratio of 0.03, and increased maximum image resolution of 1,024.

### D.4.1   Evaluation Setting

We adhere to the evaluation dataset split established in the VHM [44] and utilize their task-specific prompts for each evaluation task. To ensure fair comparison, we augment each prompt with the appropriate task identifier when evaluating models that support task identifier. For response generation across all models and tasks, we maintain consistent hyperparameters with a temperature of 1.0 and top-p of 1.0.

**Evaluation Dataset:**   (1) **Classification**: Brief introductions to each classification dataset can be found in Section D.3.1. (2) **RSVQA**[33]: RSVQA is a RS vision question answering dataset available in high resolution (RSVQA-HR) and low resolution (RSVQA-LR) versions. RSVQA-HR contains four question types: comparison (comparing object quantities), presence (determining if images contain specific objects), counting (quantifying objects), and area (measuring object areas). RSVQA-LR includes an additional rural/urban type to classify image settings. Following established practices in existing RS-specific VLMs such as VHM[44], SkysenseGPT [36], and GeoChat [27], we excluded questions related to counting and area measurement for evaluation. (3) **Grounding**: We used VG-DIOR [74] for vision grounding tasks. VG-DIOR is a RS vision grounding dataset built from RSVG [74] on the DIOR RS detection dataset [28]. Each sample contains an object description that requires the model to predict object coordinates. We calculated the IoU between predicted and ground truth coordinates, considering predictions with IoU greater than 0.5 as correct. (4) **General Knowledge**: We used LHRS-Bench [39] to holistically evaluate VLMs across different domains. LHRS-Bench is a multiple-choice question answering dataset covering 11 evaluation dimensions (from basic recognition to complex reasoning). For evaluation, we used the following prompt template: *Question. Please answer the above question with the given choice (just answer with choice index): Choices*. We extracted choice indices (e.g., A, B, C, D) using regular expressions and only considered responses containing the exact choice index as correct. Responses that included the content of the choices were not counted as correct. We did not implement the circular evaluation protocol introduced in [39].

### D.5   GRPO Training

We implement the GRPO algorithm with our ScoreRS model for rewarding open-ended question responses. We select GRPO due to its simplicity and computational efficiency.

For open-ended vision question answering tasks, we employ a binary accuracy reward: 1 if the prediction matched the ground truth, and 0 otherwise. For bounding box prediction tasks, we implement a graduated reward system based on IoU: 0.5 for IoU > 0.5, 0.6 for IoU > 0.6, 0.7 for IoU > 0.7, and 1.0 for IoU > 0.8, with 0 reward otherwise. The application of ScoreRS for close-ended questions has been previously described in Section 3.3, and the hyperparameter $\beta$ in Equation 3 is set to 0.2. We also weight the format reward at 0.3 following the practice established in R1-VL [76].

We conduct full parameters training during the GPRO RL process. During the generation phase of GRPO, we configure the sampling temperature to 0.9, and top-p to 0.99. The maximum generated tokens are limited to 4096. We set the KL penalty coefficient to 0.01 and use a batch size of 16, with each sample generating 5 candidate answers for reward calculation.

We optimize the model using AdamW with a cosine learning rate schedule and a base learning rate of $1 \times 10^{-6}$. To improve computational efficiency, we utilized bfloat16 precision, gradient accumulation with a factor of 2, and a maximum image resolution of 1024 pixels. Memory consumption is reduced

by implementing Flash Attention in conjunction with FSDP strategy [80]. All experiments are conducted for 2 episodes.

For the supervised finetuning of Qwen2VL-7B-RS with our manually collected reasoning data, we employ identical hyperparameter settings as described in Section D.4, with the LoRA rank set to 128.

**Evaluation Setting**  The evaluation metrics, datasets, and framework for assessing the trained reasoning model remained consistent with those detailed in Section D.4. The key distinction is the implementation of an answer extraction process from the reasoning model's output for evaluation purpose. Importantly, if this parsing process failed to extract a valid answer from a reasoning model, we classify the response as incorrect.

# E    Qualitative Example

## E.1    Ranked Data Comparison

In order to provide a qualitative comparison of samples receiving higher scores from different scoring models, we plotted the highest and lowest ranked samples from RemoteCLIP, VHM image-caption, and VHM vision instruction datasets in Figure 12.

From these results, we observe that for RemoteCLIP image descriptions, ScoreRS prefers more detailed descriptions while assigning lower scores to samples containing incorrect qualitative descriptions or claims that cannot be verified from the image. For the VHM image-caption dataset, ScoreRS can precisely identify accurate descriptions while filtering out incorrect or hallucinated content more effectively than CLIP-score.Finally, regarding the VHM vision instruction dataset, we clearly see that ScoreRS favors samples that demonstrate high-level reasoning and relevance to specific applications, while effectively filtering out incorrect answers.

## E.2    Qwen2VL-7B-RS-R1 Inference Example

We provide representative conversation examples generated by our finetuned model, Qwen2VL-7B-RS-R1, in Figure 13.

# F    Discussion nand Limitation

## F.1    CLIP-Score or ScoreRS?

The results in Table 1, Table 2, and Table 10 indicate that the performance improvement achieved through data filtering with ScoreRS may be somewhat marginal compared to CLIP-score (approximately 1% improvement on RemoteCLIP data, while 4% on Skyscript). However, it is important to acknowledge that the RemoteCLIP and Skyscript datasets typically contain only short, brief descriptions (e.g., *A ship in the middle of the picture*), which aligns with the type of text captions CLIP is trained on [48, 69]. Furthermore, during evaluation on classification tasks, we typically employ simple text templates such as *A photo of {class name}* as the representative embedding for class matching. The descriptions used in retrieval evaluation tests are similarly short and concise. In contrast, the preference data used to train our ScoreRS typically consists of longer, more informative text (Figure 10), which may explain why ScoreRS does not show substantial advantages when qualifying these particular datasets. Nevertheless, even in these scenarios, our ScoreRS outperforms CLIP-score, demonstrating the significant potential of using VLMs as scoring models for data qualification.

We should acknowledge that for applications and datasets similar to RemoteCLIP or Skyscript, using CLIP-score is a reasonable approach. However, for qualifying more complex vision-language data and for advanced applications such as BoN selection and reinforcement learning, ScoreRS offers superior performance, as evidenced by Table 3, Figure 4, and Table 5.

## F.2    Reasoning for Reference Based Reward Calculation

In Section 3.3, we introduce the reference-based reward mechanism using our ScoreRS for utilizing open-ended questions. Before arriving at this approach, we explored multiple strategies. The following describes our unsuccessful attempts and our reasoning:

**Direct Reward** We initially attempted to directly use the output of ScoreRS for rewarding. However, since the output of ScoreRS can be significantly larger or smaller than the 0-1 range, and we do not know the possible range of the ScoreRS output (therefore, could not normalize them), the training proved very unstable. Additionally, the rule-based reward for close-ended questions was greatly overshadowed. Although we acknowledge the potential challenges and attempt to mitigate them during the training process of our ScoreRS, the results presented in Appendix C.1 demonstrate that using raw scores for training provides the best training stability.

**Function Normalization Reward** We then try normalizing the ScoreRS output with sigmoid or other similar normalization functions (tanh). This approach was problematic because the range of our ScoreRS output likely falls in the convergence part of these functions (for example, in the $> 10$ or $<$ -10 regions of the sigmoid function). Although we find ways to sketch the domain of the function, learning remains too slow and does not yield satisfying results.

With these unsuccessful attempts behind us, we opt for the reference-based reward calculation. Although the requirement for reference datasets may reduce the possible data size for RL training, we conclude that data volume is not the critical part of RL training—better answer sampling and reward strategies are far more important. Of course, we do not consider this the optimal solution, but rather our current finding, and we hope future work will explore more effective approaches.

### F.3 RS Vision-Language Data Quality

In this work, we have developed a framework to evaluate the quality of RS vision-language data and demonstrated the effectiveness of parameter-wise learned scoring models for RS vision-language data selection. Our investigation reveals that current RS vision-language datasets fall considerably short of optimal quality, underscoring the need for more rigorous curation efforts in this domain-specific context.

As the machine learning community increasingly validates the "less is more" principle [25, 82, 72], the RS community should prioritize data quality improvement through systematic quality control mechanisms or scoring model implementations. While our work highlights the suboptimal quality of existing RS vision-language data using our ScoreRS, we acknowledge that ScoreRS is merely an initial attempt to qualify such data. Future research should investigate whether using separate scoring models for each quality dimension introduced in Appendix A could bring additional benefits. Further exploration should also address various dimensions including data difficulty levels, category distributions, and optimal data combinations to fully harness the potential of RS-specialized VLMs.

### F.4 Expectation for Building RS-Specific VLMs

Throughout our investigation and evaluation, we gained significant insights regarding the misalignment between current data characteristics and the capabilities expected from advanced VLMs.

Modern large-scale VLMs excel not only in perception tasks but also in reasoning and planning for complex problem-solving [81]. For the RS community, these models should ideally support sophisticated applications such as disaster analysis, urban planning, and transportation system assessment. Our analysis suggests that current RS-specific VLMs, while performing admirably on basic perception tasks, fall short on more complex challenges compared to general VLMs (Table 4).

Since basic perception tasks can be effectively addressed with more lightweight models (for example, accuracy on AID can achieve over 95% with the much smaller ViT-B architecture [60]), the development of large RS-specific VLMs could, in our view, put more focus on expanding beyond these foundational capabilities toward building agent-like assistants that interact with RS experts to conduct geological information analysis. This requires equipping models with more advanced capabilities such as planning, task decomposition, and reflection. Therefore, while continuing to value and build upon the important work in perception tasks, we suggest that future RS-specific VLM development could benefit from increased emphasis on these more complex reasoning capabilities to fully leverage the potential of LLMs in RS applications.

We advocate for the development of VLMs that transcend basic perception tasks and deliver advanced analytical capabilities specific to RS applications. Furthermore, we hypothesize that exposure to more challenging, application-oriented RS analysis tasks could paradoxically enhance these models'

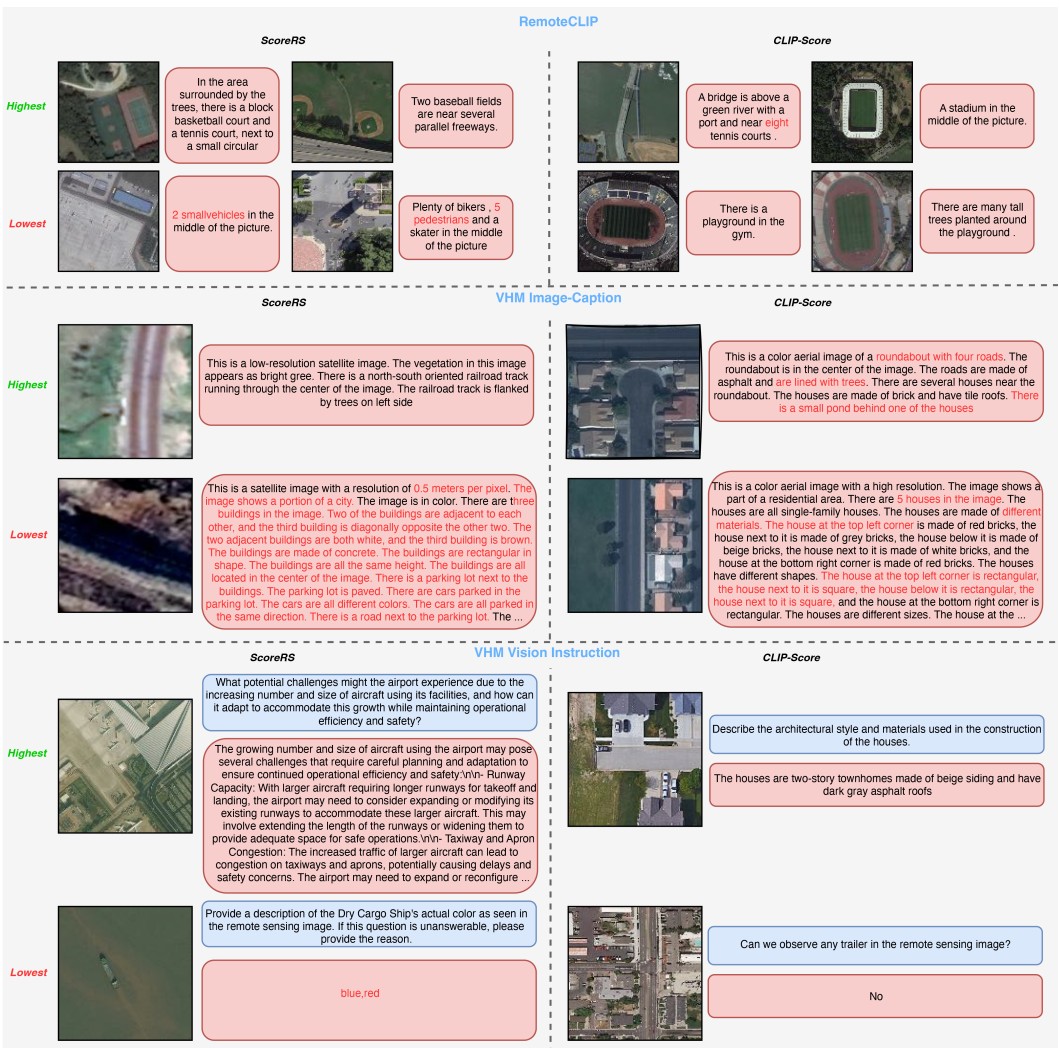

Figure 12: Ranked examples from different dataset with different score models. Red represents low-quality expression

fundamental perception capabilities [72]. The creation of high-quality, application-specific datasets represents a critical direction for future work in this domain.

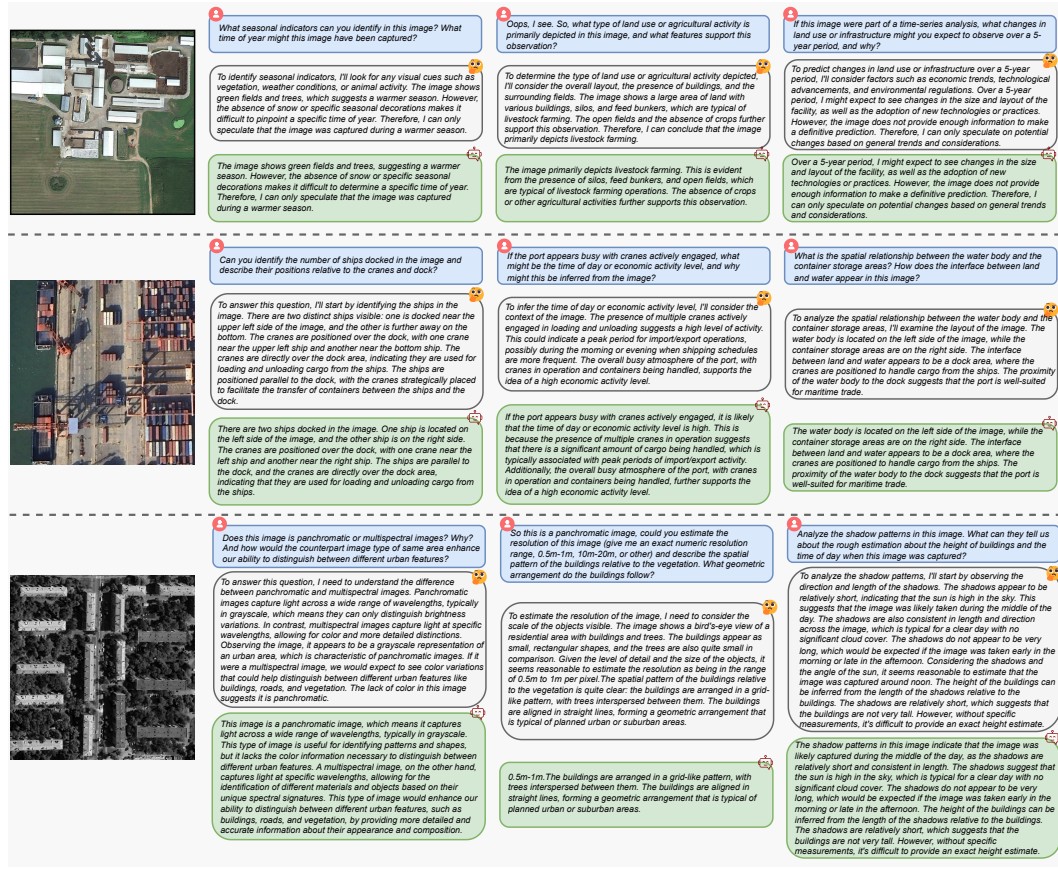

Figure 13: Representative conversation examples demonstrating the capabilities of our finetuned Qwen2VL-7B-RS-R1 model

Table 16: Scoring system for RS image-caption

| Aspect | Score | Definition for Descriptions/Captions |
|---|---|---|
| 1. Relevance | 1 (Poor) | The description is largely irrelevant to the image, contains significant hallucinations (objects/features clearly not present), or fundamentally misrepresents the scene. |
| | 2 (Weak) | The description has notable inaccuracies, some hallucinated minor elements, or only vaguely relates to the primary content of the image. |
| | 3 (Fair) | The description is generally relevant and mostly accurate, but may contain minor inaccuracies, slight misinterpretations, or omit verification of some mentioned details. |
| | 4 (Good) | The description is accurate, clearly relevant, and faithfully represents the main visual content with no significant hallucinations or misinterpretations. Minor, non-critical details might be simplified. |
| | 5 (Excellent) | The description is perfectly accurate, highly relevant, and precisely reflects all key visual elements and their relationships in the image without any ambiguity or hallucination. |
| 2. Specificity & Detail | 1 (Poor) | The description is extremely vague (e.g., "an image of the ground," "some features") and provides almost no useful detail about the RS image content. |
| | 2 (Weak) | The description is mostly general, lacking specific details about objects, patterns, or spatial arrangements that are clearly visible. |
| | 3 (Fair) | The description provides some specific details but remains general in many aspects. Key distinguishing features might be mentioned but not elaborated upon. |
| | 4 (Good) | The description offers a good level of specific detail about important objects, their characteristics (if discernible), and their general layout. |
| | 5 (Excellent) | The description is rich in specific, fine-grained details about objects, textures, shapes, counts (where appropriate), and their precise spatial relationships. |
| 3. Completeness | 1 (Poor) | The description misses almost all salient and important features/regions in the RS image. |
| | 2 (Weak) | The description omits several significant features or covers only a small, non-representative part of the image. |
| | 3 (Fair) | The description covers some of the main features but misses other notable ones or lacks comprehensive coverage of the overall scene. |
| | 4 (Good) | The description covers most of the salient features and important regions of the image adequately. |
| | 5 (Excellent) | The description provides a comprehensive account of all major salient features, land cover types, and significant patterns visible across the entire image. |
| 4. Clarity & Fluency | 1 (Poor) | The description is largely incomprehensible, grammatically incorrect, or uses language so poorly that its meaning is lost. |
| | 2 (Weak) | The description is difficult to understand due to significant grammatical errors, awkward phrasing, or unclear language. |
| | 3 (Fair) | The description is mostly understandable but contains some grammatical errors, awkward phrasing, or minor ambiguities. |
| | 4 (Good) | The description is clear, grammatically correct, and well-phrased, making it easy to understand. |
| | 5 (Excellent) | The description is exceptionally clear, concise, grammatically flawless, and uses precise, fluent language. |
| 5. Semantic Richness | 1 (Poor) | The description uses no relevant remote sensing terminology or misuses terms completely. Lacks any understanding of RS-specific concepts. |
| | 2 (Weak) | The description uses very generic terms (e.g., "green areas," "buildings") with minimal or incorrect RS-specific vocabulary. |
| | 3 (Fair) | The description uses some basic and appropriate RS terminology (e.g., "urban area," "farmland") but lacks depth or precision in describing RS-specific features. |
| | 4 (Good) | The description correctly employs relevant RS terminology to describe features, land cover, or patterns (e.g., "center-pivot irrigation," "industrial zone"). |
| | 5 (Excellent) | The description expertly uses precise and advanced RS terminology, accurately identifying and describing complex features, phenomena, or sensor characteristics relevant to the image. |

Table 17: Scoring system for RS instructions samples (Part 1)

| Aspect | Score | Definition for Instruction/QA Pairs |
|---|---|---|
| **1. Relevance** | **1 (Poor)** | The answer is completely irrelevant to the question, is based entirely on hallucinated information not in the image, or confidently answers an unanswerable question incorrectly. |
| | **2 (Weak)** | The answer poorly addresses the question, contains significant inaccuracies based on the image, or makes large, unfounded assumptions. Wrongly asserts unanswerable question is answerable. |
| | **3 (Fair)** | The answer attempts to address the question and is mostly based on the image, but may have minor inaccuracies, misinterpretations, or if the question is unanswerable, it may fail to state this clearly or attempt a guess. |
| | **4 (Good)** | The answer accurately and directly addresses the question using information verifiable in the image. If the question is unanswerable from the image, it states so clearly. |
| | **5 (Excellent)** | The answer perfectly and precisely addresses all aspects of the question using only visual evidence from the image. If unanswerable, it explicitly and correctly states why based on image content. |
| **2. Specificity & Detail** | **1 (Poor)** | The answer is extremely vague (e.g., "Yes," "Maybe," "Objects are present") and provides no specific information from the image relevant to the question. |
| | **2 (Weak)** | The answer is too general and lacks specific details that are visible in the image and pertinent to answering the question effectively. |
| | **3 (Fair)** | The answer provides some relevant detail but could be more specific or elaborate further based on visible image content and the question's needs. |
| | **4 (Good)** | The answer provides a good level of specific detail directly from the image that sufficiently addresses the question. |
| | **5 (Excellent)** | The answer is highly specific and rich in relevant details extracted from the image, providing a precise and thorough response to the question. |
| **3. Completeness** | **1 (Poor)** | The answer completely fails to address the core of the question or ignores key components of a multi-part question. |
| | **2 (Weak)** | The answer addresses only a small part of the question or provides a very superficial response, missing obvious follow-up details implied by the question and visible in the image. |
| | **3 (Fair)** | The answer addresses the main part of the question but may be incomplete, missing some nuances, or not fully utilizing available visual information. If question is partially unanswerable, this might not be fully clarified. |
| | **4 (Good)** | The answer comprehensively addresses all explicit parts of the question using available image information. Clearly indicates if parts are unanswerable. |
| | **5 (Excellent)** | The answer fully and exhaustively addresses all aspects of the question, including implicit sub-questions where appropriate, based on thorough image interpretation. Clearly delineates what can and cannot be answered. |
| **4. Clarity & Fluency** | **1 (Poor)** | The question is incomprehensible, or the answer is grammatically nonsensical, making the entire QA pair useless. |
| | **2 (Weak)** | The question is ambiguous, or the answer is poorly phrased with significant grammatical errors, making the QA pair difficult to understand or trust. |
| | **3 (Fair)** | The question is mostly clear, and the answer is generally understandable but may contain minor grammatical errors, awkward phrasing, or slight ambiguities. |
| | **4 (Good)** | The question is clear and unambiguous. The answer is well-phrased, grammatically correct, and easy to understand. |
| | **5 (Excellent)** | The question is exceptionally clear and well-posed. The answer is perfectly fluent, concise, grammatically flawless, and directly responsive. |

Table 18: Scoring system for RS instructions samples (Part 2)

| Aspect | Score | Definition for Instruction/QA Pairs |
|---|---|---|
| **5. Semantic Richness** | **1 (Poor)** | Neither the question nor the answer uses any relevant RS terminology, or terms are severely misused, showing no domain understanding. |
| | **2 (Weak)** | The QA pair uses very generic terms. If RS terms are used, they are minimal, overly simplistic for the context, or slightly incorrect. |
| | **3 (Fair)** | The question or answer uses some basic, appropriate RS terminology, but could be more precise or leverage more domain-specific knowledge relevant to the image. |
| | **4 (Good)** | The question and/or answer correctly employ relevant RS terminology that enhances the specificity and technical accuracy of the exchange. |
| | **5 (Excellent)** | The QA pair expertly uses precise and potentially advanced RS terminology. The question might probe RS-specific insights, and the answer provides them accurately. |

Table 19: Prompts for selecting image-caption preference pair

System Message:
You are a highly precise and analytical image description evaluator. Your task is to select the most accurate caption from given options or provide your own if none are satisfactory. You should approach this task systematically and provide detailed justification while maintaining objectivity.

Prompt:
Please analyze the given image and evaluate these three captions:
Caption 1: {caption1}
Caption 2: {caption2}
Caption 3: {caption3}
Follow these steps:
1. First, provide your own detailed description of the image, focusing on:
- Observable objects and their characteristics
- Spatial relationships and positioning
- Key visual patterns and structures
- Colors and textures
Avoid making assumptions about image type or source.
2. Evaluate each provided caption using the following criteria:
The ceiteria in Table 16.
3. Provide your analysis in the following format:
[description]
your independent description
[end-description]
[evaluation]
1: [score],[score],[score],[score],[score]
2: [score],[score],[score],[score],[score]
3: [score],[score],[score],[score],[score]
[end-evaluation]
[selection]
best-id: [id]
best-reason: [reason]
[end-selection]

Table 20: Prompts for selecting vision instruction preference pair

You are an expert evaluator specialized in determining the quality of answers. Your task is to systematically analyze and compare answers to determine which one is better. You must:
1. Always analyze the core requirements of the question first
2. Break down both the given answers into key components
3. Compare them systematically and objectively based on the given criteria
4. Provide conclusions in a strictly formatted output and
Your evaluation must be based on the following criteria:
The criteria in Table 17 and Table 18
Now, please evaluate the following answer pair:
QUESTION:{question}
ANSWER:
{answer1}
{answer2}
{answer3}
Provide your evaluation with the following format:
[evaluation]
1: [score],[score],[score],[score],[score]
2: [score],[score],[score],[score],[score]
3: [score],[score],[score],[score],[score]
[end-evaluation]
[selection]
best-id: [id]
best-reason: [reason]
[end-selection]

Table 21: Manually designed questions for RS specific applications (Part 1)

Basic Visual Recognition
What is the dominant land cover type in this image?
How many distinct types of land cover can you identify in this image?
What percentage of the image is covered by water bodies?
Are there any clouds present in the image? If yes, approximately what percentage of the image is cloud-covered?
What season does this image appear to be taken in? What visual cues support your answer?

Spatial Analysis
What is the approximate scale of this image? (city-scale/regional/continental)
Describe the spatial distribution of urban areas in relation to natural features.
What patterns of human settlement can you observe? (clustered/dispersed/linear)
How does the terrain influence the distribution of vegetation?
Can you identify any transportation networks? How do they relate to urban development?

Environmental Assessment
Are there any visible signs of environmental degradation?
Can you identify potential areas of soil erosion?
What evidence of water pollution can you observe?
How healthy does the vegetation appear? What indicators are you using?
Are there any visible impacts of climate change or extreme weather events?

Urban Analysis
What is the predominant urban development pattern?
Can you identify different types of urban land use (residential/commercial/industrial)?
How well-connected is the transportation infrastructure?
Are there clear boundaries between urban and rural areas?
Can you identify any informal settlements or rapid urbanization patterns?

Agricultural Analysis
What types of agricultural practices are visible in the image?
How does field size and shape vary across the image?
Can you identify any irrigation systems or water management features?
What is the current stage of crop growth in the visible fields?
Are there any visible patterns of crop rotation or fallow land?

Disaster Assessment
What evidence of natural disasters can you observe?
How has infrastructure been affected by the disaster?
Can you identify areas at risk of future disasters?
What emergency response activities are visible?
How has the landscape changed post-disaster?

Geological Features
What major geological formations are visible?
Can you identify any fault lines or tectonic features?
What types of erosional patterns are present?
Are there any visible mining or extraction activities?
How does geology influence vegetation patterns?

Infrastructure Analysis
What types of energy infrastructure can you identify?
How well-developed is the transportation network?
Can you locate major water management facilities?
What patterns of industrial development are visible?
How does infrastructure density vary across the image?

Temporal Understanding
What time of day was this image captured? What shadows or lighting conditions support your answer?
Which season is represented in this image? What evidence supports this?
What indications of recent urban development can you identify?
What stage of vegetation growth is visible in different areas?
What evidence of water level fluctuations can be observed from shoreline features?

Table 22: Manually designed questions for RS specific applications (Part 2)

Advanced Reasoning
Based on the visible patterns, what are the main economic activities in this region?
How do natural features constrain or enable human development?
What ecosystem services are visible in this image?
How sustainable are the visible land use practices?
What future development challenges might this area face based on current patterns?
Quantitative Assessment
What is the approximate area covered by different land use types?
How dense is the urban development in different parts of the image?
What is the ratio of built-up area to green space?
How fragmented are the natural habitats?
What is the distribution pattern of settlement sizes?
Color Spectral Analysis
What does the variation in vegetation color indicate about plant health?
Can you identify areas of bare soil based on color signatures?
What do the color patterns in urban areas suggest about building materials?
How do seasonal changes affect the spectral signatures of different features?

Table 23: Instruction for prompting models to answer the RS specific questions

You are an expert remote sensing analyst. Examine the provided satellite image and answer the given question.
Remember to:
1. Start with direct observations
2. Base all conclusions on visible evidence only
If you cannot answer the question with the available information, please explicitly state what cannot be determined and explain specifically what prevents you from doing so.

Table 24: Instruction for prompting models to rephrase the questions

You are an expert in remote sensing and satellite image analysis. Transform the following question into a new, more diverse version. The new question should:
1. Test the same core concept but approach it differently
2. Either increase or decrease the complexity
3. Change the context or application domain
4. Use a different response format or analytical approach
Also please task care that there is just single image. So do not output any temporal question. Moreover, do not explicitly mention the image is satellite image.
Original question: "{question}"
Generate ONE new question that maintains the spirit of the original but differs in at least 2 of the above aspects. Do not explain your choices - only output the new question.
Output here:

