# OpenReview forum: "Quality-Driven Curation of Remote Sensing Vision-Language Data via Learned Scoring Models"
_NeurIPS.cc/2025/Conference — NeurIPS 2025 poster_

### Official Review · Reviewer_EhZY · 2025-06-30

**Clarity:** 4
**Significance:** 3
**Originality:** 3
**Rating:** 5
**Confidence:** 4

**Summary:**

Many existing vision-language (VL) datasets in remote sensing (RS) rely on automatic or semi-automatic curation pipelines, often lacking rigorous quality control. This work introduces a scoring model trained on large-scale RS vision-language preference data for automated quality assessment. Extensive experiments demonstrate the effectiveness of the proposed scoring method for data filtering.

**Questions:**

See Weaknesses above.

**Ethical Concerns:**

["NO or VERY MINOR ethics concerns only"]

**Final Justification:**

Most of the concerns have been addressed and I decide to keep my original score.

**Limitations:**

Limitations regarding the quality of preference data should be discussed.

**Quality:**

3

**Strengths And Weaknesses:**

Strengths:
1)Data quality is a critical factor in developing more powerful and reliable VL models (VLMs) in RS. This work provides quantitative evidence that current RS vision-language datasets are suboptimal and require improved quality control measures.
2)The experimental validation is thorough, with sufficient implementation details provided in the supplementary materials. The quantitative results clearly support the effectiveness of the proposed scoring method for data filtering.

Weaknesses
1)The claim that *"studies such as VHM… leverage flagship VLMs like GPT-4 or Gemini for synthetic vision-language data curation"* is questionable. Most referenced works primarily use rule-based techniques to construct their instruction-following datasets. In contrast, recent works like VRSBench[1] and ChatEarthNet[2] explicitly employ VLMs for vision-language data curation. This distinction should be clarified.
2)RSGPT[3] was released before GeoChat and stands as one of the pioneering VLMs in RS. Its contribution should be acknowledged in the related work section for a more complete literature review. Moreover, recent surveys on this topic could be included.
3)In Section 3.2.1, the proposed preference dataset relies on GPT-4o as the scoring system. While this automation streamlines data curation, the reliability of GPT-4o as a preference judge remains uncertain. Given that the dataset contains 76K image-text preference pairs, evaluating only 1,000 random samples may not sufficiently validate GPT-4o’s consistency and bias. Additionally, GPT-4o might have been trained on limited RS-specific data, introducing potential biases. The same concern applies to Section 3.2.2, where *Qwen2VL-72B* is used as a preference judge.

[1] Li, X., Ding, J., & Elhoseiny, M. (2024). Vrsbench: A versatile vision-language benchmark dataset for remote sensing image understanding. arXiv preprint arXiv:2406.12384.
[2] Yuan, Z., Xiong, Z., Mou, L., & Zhu, X. X. (2024). Chatearthnet: A global-scale image-text dataset empowering vision-language geo-foundation models. Earth System Science Data Discussions, 2024, 1-24.
[3] Hu, Y., Yuan, J., Wen, C., Lu, X., Liu, Y., & Li, X. (2025). Rsgpt: A remote sensing vision language model and benchmark. ISPRS Journal of Photogrammetry and Remote Sensing, 224, 272-286.

---

> ### Author Rebuttal · Authors · 2025-07-29
>
> **We truly appreciate the reviewer's recognition of our method's importance in demonstrating the necessity for improved quality control in remote sensing vision-language data, our comprehensive validation, and sufficient implementation details. Regarding concerns about further discussion of the proposed preference data quality and related work review, our responses are as follows:**
>
> **Q1: Regarding the possible bias and accuracy of using GPT-4o or Qwen2VL-72B as judging models for preference dataset**
>
> **A1:** We acknowledge potential accuracy concerns when using GPT-4o and Qwen2VL-72B for preference judgment. However, we compared these models against domain-specific VLM models as preference judges. Since current large VLMs in remote sensing are only 13B parameters, not fine-tuned for scoring different answers with given criteria, and trained on potentially low-quality data, their instruction-following capabilities as judging models with our scoring system are inferior to GPT-4o or Qwen2VL-72B (in our human validation: LHRS-Bot-Nova 82.8% vs. GPT-4o 92.6%). Although GPT-4o and Qwen2VL-72B have limited RS data exposure, their training on more diverse data distributions and instruction datasets provides sufficient instruction-following and context-dependent answering capabilities for this task.
>
> To further validate our preference dataset accuracy, we selected 3 groups of preference data from image-caption (IC) and vision-instruction (VI) preference datasets (1K samples each: 500 from ICPD, 500 from VIPD). These 3K samples differ from the manuscript validation experiment. Three human annotators judged whether chosen positive samples were better than negative samples (Yes or No). We computed accuracy to validate GPT-4o and Qwen2VL-72B as judges. Additionally, the 3 groups shared 100 common samples (50 from ICPD, 50 from VIPD) to validate inter-annotator agreement.
>
> |  Accuracy @ ICPD   |  Accuracy @ VIPD   | Human Consistency @ ICPD | Human Consistency @ VIPD |
> | :----------------: | :----------------: | :----------------------: | :----------------------: |
> | 90.06% (1351/1500) | 96.80% (1452/1500) |       88% (44/50)        |        96%(48/50)        |
>
> Results show that both GPT-4o (ICPD) and Qwen2VL-72B (mostly VIPD) align well with human judgment. However, image captions are lengthy and difficult to quantify, leading to varying human preferences.
>
> We acknowledge this curation strategy is not optimal. Future work could incorporate more complex construction methods, such as RAG with human-annotated judges as data sources and self-reflection mechanisms to improve judgment results.
>
>
>
> **Q2: Regarding more holistic literature review and clarification about data construction**
>
> **A2:** Thank you for your valuable suggestions. In our early discussion of remote sensing vision-language dataset construction, we distinguished different methods based solely on their image-caption datasets, which neglected vision-instruction datasets. We will revise and better clarify these differences. Additionally, we will include more comprehensive coverage of remote sensing vision-language models, their timeline, and relevant surveys.

---

> > ### Comment · Reviewer_EhZY · 2025-08-05
> >
> > I'm mostly happy with author's response and will keep my score.

---

### Official Review · Reviewer_3m9H · 2025-07-02

**Clarity:** 3
**Significance:** 3
**Originality:** 2
**Rating:** 4
**Confidence:** 4

**Summary:**

This paper presents ScoreRS, a learned scoring model for assessing the quality of remote sensing vision-language data. ScoreRS is trained on a large-scale RS vision-language preference dataset, which will be made publicly available. The model is trained via a standard three-stage training strategy, and is designed to assess the best vision-language pairs across five quality dimensions defined by the authors. ScoreRS is applied to filter data for CLIP fine-tuning and is also used for reinforcement learning and as a Best-of-N (BoN) selector at test time for VLMs. The authors demonstrate improvements over CLIP-score filtering and claim performance gains when utilizing the top-30% of data ranked by ScoreRS.

**Questions:**

Authors are encouraged to include more RS-specific baselines as scoring models for data filtering (see Weaknesses[1]). Such baselines, if incorporated in Tables 1-2, would significantly strengthen the paper's comparative analysis and clarify whether ScoreRS offers a real advantage. For example, SkyCLIP [2] provides a filtered dataset based on CLIP-LAION-RS scoring, which seems directly comparable to the proposed ScoreRS approach.

Composed image retrieval task (C2I) can be added as a column in Table 2. Prior work [1] has already evaluated CLIP and RemoteCLIP using WeiCom on top of these pre-trained models on the PatternCom dataset. The last two rows of Table 2 can be evaluated similarly.

Beyond RS-specific models, more recent, gereral-purpose alternatives to CLIP-Score, such as SigLIP/SigLIP-2, could be considered as filtering baselines. If any of these experiments are not applicable for technical or other reasons, can the authors clarify why they are not applicable in the proposed pipeline?

Table 1 and Table 2 could benefit by including additional pre-trained models in the evaluation, such as those provided by SkyCLIP [2].

Potential bias in RL. In Reinforcement Learning experiments, ScoreRS is used as a scoring model for data filtering (for the RL-trained models) and then as a reward model. This raises concerns about potential evaluation bias. Can the authors clarify what steps where taken to mitigate this bias optimism toward ScoreRS method?

Computational cost. Could the authors provide the training costs (GPU hours or FLOPs) for each stage of ScoreRS training, downstream fine-tuning and the inference runtime using ScoreRS? This is important for efficiency and feasibility assessments.

[1] Psomas, B., Kakogeorgiou, I., Efthymiadis, N., Tolias, G., Chum, O., Avrithis, Y., & Karantzalos, K. (2024). Composed image retrieval for remote sensing. In IGARSS 2024 – IEEE International Geoscience and Remote Sensing Symposium.
[2] Wang, Z., Prabha, R., Huang, T., Wu, J., & Rajagopal, R. (2024, March). Skyscript: A large and semantically diverse vision-language dataset for remote sensing. In Proceedings of the AAAI Conference on Artificial Intelligence (Vol. 38, No. 6, pp. 5805-5813).

**Ethical Concerns:**

["NO or VERY MINOR ethics concerns only"]

**Final Justification:**

I acknowledge the authors’ rebuttal and have considered the other reviewers’ comments. The authors managed to address at least partially my concerns. Still the contribution remains marginal for NeurIPS. Nevertheless, I will adjust my rating to a borderline accept.

**Limitations:**

yes

**Quality:**

3

**Strengths And Weaknesses:**

Strengths
ScoreRS address an important challenge for the remote sensing community, assessing the quality of large-scale vision-language data.

Authors evaluate the proposed ScoreRS methodology across various tasks and applications.

The paper is well-written and the pipeline easy to follow, which makes reproducibility feasible.


Weaknesses
The comparison in Tables 1 and 2 is based only on CLIP-Score filtering and omits data filtering baselines using RS-specific models such as CLIP-RS-LAION, RemoteCLIP, or SkyCLIP (SkyCLIP provides a filtered version of its initial dataset using CLIP-LAION-RS as a scoring model). This makes it difficult to assess whether ScoreRS offers meaningful improvements over existing domain-adapted alternatives.

Also, newer models such as SigLIP and SigLIP-2 could be explored as alternatives to CLIP-Score.

The baseline comparisons are insufficient. Table 1 and Table 2 only evaluate filtering effectiveness using the generic CLIP model and RemoteCLIP trained on the full dataset. SkyCLIP’s pre-trained models, which are publicly available, could have been directly included.
ScoreRS is used both as a filter and as an evaluator (in RL experiments), introducing potential optimism bias in favor of ScoreRS.
In line 172 the authors state that they use a subset of SkysenseGPT (381K samples). It’s unclear how this subset was filtered.
Typographical Error in Line 753: The sentence "We provide serveal samples..." should be corrected to “We provide several samples…”

Overall, while the framework has potential, is well-executed, and addresses a critical challenge for the remote sensing community, the methodology is primarily engineering-driven, and lacks sufficient novelty. Beyond that, the most crucial concern is the limited comparisons, both in terms of the baselines utilized as scoring models and the pre-trained models evaluated in the downstream tasks.

---

> ### Author Rebuttal · Authors · 2025-07-29
>
> **We sincerely appreciate the reviewer's acknowledgment of our contribution to data curation challenges in the remote sensing community, our well-written and easy-to-follow pipeline, and extensive experiments. Regarding the reviewer's concerns about more diverse comparisons, computational costs, and novelty, our responses are provided below:**
>
>
>
> **Q1: Comparison with remote-sensing specific CLIP models and SigLIP for data filtering**
>
> **A1:** Thank you for your valuable concerns. We aimed to make our comparisons as solid and extensive as possible, and we initially considered which CLIP models to use as baselines. We chose default CLIP based on two key reasons:
>
> 1. **Length extension capabilities**: Most RS-specific CLIP models (RemoteCLIP, SkyCLIP) were fine-tuned on datasets with short captions. We wanted to compare ScoreRS not only in CLIP fine-tuning scenarios but also in large vision-language fine-tuning scenarios where captions and multi-turn conversations can be extremely long (such as the VHM dataset we used to fine-tune Qwen2VL).
>
>     CLIP has its extension version Long-CLIP [1] (although we acknowledge it is not official), while SigCLIP and SigLIP-2 have a maximum length of 64 tokens, and other RS-specific CLIP models are limited to 77 tokens (same as CLIP).
>
> 2. **Trustworthiness of fine-tuned versions**: Since we demonstrated that current remote sensing vision-language data is relatively low-quality, we were concerned that fine-tuned versions might be biased toward this low-quality data, resulting in inferior performance.
>
> Following your concerns, we noted that CLIP-LAION-RS was fine-tuned on an RS subset of the well-curated LAION-2B dataset, and its filtered version outperforms the CLIP-filtered version. Therefore, we curated the RemoteCLIP dataset using both SigCLIP-2 (L/16-256) and CLIP-LAION-RS (L/14), comparing results with our ScoreRS for fine-tuning CLIP (we did not compare LVLM fine-tuning due to reason 1). Results are listed below:
>
> ( Saving Ratio=30%, Classification Dataset={NWPU, AID, METERML, WHU-RS19, SIRI-WHU, fMoW, EuroSAT}, Retrievel Dataset={UCM, RSICD} )
>
> | Method        | Classification Avg. @ 1 | Retrieval Avg @ 1 |
> | ------------- | :---------------------: | :---------------: |
> | ScoreRS       |          70.97          |       27.11       |
> | CLIP-Score    |          69.98          |       26.36       |
> | SigCLIP-2     |          68.24          |       26.47       |
> | CLIP-LAION-RS |          70.59          |       26.41       |
>
> From the results, although SigCLIP-2 shows better performance on various tasks in general compared to CLIP [2], factors such as data distribution and training recipes may all contribute to performance differences in the remote sensing domain. Therefore, the SigCLIP-2 filtered version shows inferior performance on classification tasks but improvement on retrieval compared to CLIP. On the other hand, the CLIP-LAION-RS filtered version shows better performance on both classification and retrieval tasks compared to CLIP. Nevertheless, our ScoreRS filter demonstrates the best performance among all these baselines.
>
> From these results and other extensive experiments, it is clear that although our ScoreRS is larger than CLIP and its variants, it is the first score model in the RS community that can be used to evaluate different types of vision-language data. Additionally, thanks to the strong capabilities of the Qwen2VL backbone, it can even be used to evaluate multi-image vision-language data such as change captions for multi-image.
>
> [1]  Long-CLIP: Unlocking the Long-Text Capability of CLIP. Zhang et. al., ECCV2024.
>
> [2] SigLIP 2: Multilingual Vision-Language Encoders with Improved Semantic Understanding, Localization, and Dense Features. Tschannen et. al., arXiv:2502.14786.
>
>
>
> **Q2: Computational cost of our pipeline**
>
> **A2:** We provide detailed computational costs for each component. ScoreRS training uses 8×A100 GPUs with Flash Attention and ZeRO-2 optimization, requiring approximately 2 hours for stage 1, 10 hours for stage 2, and 18 hours for stage 3. Score inference achieves ~1.4 batches/second on A100 with batch size 8, maximum sequence length 4096, Flash Attention, and FP16 precision. Downstream fine-tuning of Qwen2-VL-7B uses 8×A100 GPUs with LLaMA Factory optimizations, taking ~10 hours for stage 1 and ~6 hours for stage 2. RL training employs two nodes with 8×A100 each (one for serving 8 ScoreRS reward models, another for policy training) and completes within 8 hours using VeRL VLLM pipeline optimization.
>
>
>
> **Q3: The methodology is primarily engineering-driven, and lacks sufficient novelty**
>
> **A3:** We appreciate this concern and acknowledge that our work is primarily engineering-driven. However, as you mentioned, we have built a complete framework addressing a critical challenge for the remote sensing community, demonstrating whether, how, and what approaches work effectively in the RS domain and plan to open-source our efforts to benefit the broader community. Also, we believe our preference data construction pipeline and RL strategy offer novel contributions compared to existing work.
>
> In the current paradigm where engineering and research increasingly work hand-in-hand [2], [3], while novel methods remain essential for community advancement, demonstrating how to make existing paradigms more powerful for specific domains also constitutes valuable contribution.
>
> [2] On the opportunities and risks of foundation models. Bommasani et. al. arxiv:2108.07258, 2021.
>
> [3] A survey of large language models. Zhao et. al. arxiv:2303.18223, 2023.
>
>
>
> **Q4: Question about the source of SkysenseGPT subset**
>
> **A4:** We did not filter this subset ourselves, but directly used the subset provided by the official SkysenseGPT dataset on Hugging Face.
>
>
>
> **Q5: Potential bias introduced from using ScoreRS in both data filtering and RL process**
>
> **A5:** We acknowledge that since ScoreRS is trained on our preference dataset, any bias in the preference dataset propagates to ScoreRS and subsequently to both data filtering and RL training processes. While we recognize this bias is not optimal, our experimental results (Tables 1-5 and other relevant findings) demonstrate its effectiveness.
>
> It's important to note that both data quality definitions and RL processes for truthfulness, completeness, and helpfulness inherently reflect human biases about what constitutes good data quality and helpful responses. We constructed our preference dataset based on our scoring system to define good data (our bias), trained ScoreRS on this data, and ScoreRS then applied this bias to filtering and RL processes—ultimately proving beneficial.
>
> We believe that data filtering (beyond basic deduplication and grammar checking) and alignment training inherently involve models learning human biases, which are not necessarily harmful. While we acknowledge our introduced bias is not optimal, we expect future work to build upon and improve our approach.

---

> > ### Author Response · Authors · 2025-08-07
> > **Thank you for these suggestions.**
> >
> > Thank you for your valuable feedback and suggestions. Hope that we have carefully addressed all the concerns raised. We would greatly appreciate your response and any additional guidance you might have regarding our response. Thank you again!

---

### Official Review · Reviewer_5qQo · 2025-07-03

**Clarity:** 4
**Significance:** 3
**Originality:** 4
**Rating:** 5
**Confidence:** 4

**Summary:**

The paper proposes a method to score image-caption / image-instruction pairs for a remote sensing dataset. It first collects RS images and then asks various VLMs to generate captions or answer questions for instructions. It then uses GPT-4o to rank these answers on 5 different axes. The best answer/caption is chosen as a positive (and the rest as negative) sample to train a scoring function. The model trained a scoring VLM in multiple stages. This scorer can be used to filter out data to find more useful subsets of large training datasets, as the paper shows, the scorer results in significantly better fine-tuning of CLIP on remote sensing data. Moreover, the paper also shows that the scorer can be used as a reward function for RL training VLMs as well as test-time best answer selection.

**Questions:**

See weakness.

**Ethical Concerns:**

["NO or VERY MINOR ethics concerns only"]

**Final Justification:**

The authors have resolved the weaknesses and promised to add the new qualitative results. I'm upgrading the rating to an accept.

**Limitations:**

Yes

**Quality:**

3

**Strengths And Weaknesses:**

# Strengths
* The paper is very well-written and easy to read.
* The results are very comprehensive and show the usefulness of ScoreRS in various tasks and for various models in remote sensing.
* The problem motivated by the paper is very novel in the space of remote sensing imagery.
* The method for collecting the dataset and training the scorer is novel and interesting.

# Weaknesses
* The image caption/answer generation, as well as judgments on those responses, are both fully automated and done via automated VLMs.
    * While the human study shows that humans also select the same response, it is unclear if these responses are the best. i.e., both humans and GPT-4o are still ranking one out of the possible 3 outcomes to be positive. In theory, it is possible that neither of them are actually good enough. As a result, the model cannot perform better than the best of these three models used for generation.
    * In other words, the scorer is learning from the biases of existing RS-VLMs and general-purpose VLMs. If GPT-4o is good at scoring and understanding remote sensing images/captions/instructions, why do we even need remote sensing VLMs?
    * Overall, using a training scorer using automatic VLMs to filter data to fine-tune the CLIP model seems a bit circular.

## Not weaknesses per se, but comments
* Figure 3 could be extended beyond 30% to see if the performance improves further.
* In discussion from line 30-34. Alongside these three methods for obtaining data for training RS VLMs, another set of approaches that could be cited is getting supervisory signals from auxiliary data like internet images [1]
 * Figure 1, could explain in a little bit more detail what each of those terms means. The appendix has this but could be useful in the main paper too.


[1] Remote sensing vision-language foundation models without annotations via ground remote
alignment., Mall et. al. ICLR, 2024

---

> ### Author Rebuttal · Authors · 2025-07-29
>
> **Thank you for acknowledging our method's novelty, comprehensive validation, and well-written presentation. We appreciate your concerns about our preference dataset construction and provide our response below:**
>
>
>
> **Q1: The candidate responses for generating preference datasets may not be good enough.**
>
> **A1:** Excellent question—this was indeed our primary concern during dataset design. We adopted this approach because even when all responses are suboptimal, relative quality differences still exist (e.g., "not so bad" vs. "bad" vs. "terrible"). Learning to rank relatively better responses demonstrates meaningful quality judgment ability.
>
> In practice, when fine-tuning vision-language models, we cannot guarantee all training data is high-quality. In scenarios with limited resources to rebuild datasets, ScoreRS can still provide value by identifying relatively better samples, even if the absolute improvement is modest compared to improving the data itself.
>
> We acknowledge that ideal candidate generation should include more diverse models, sampling strategies, and human-in-the-loop verification—approaches used by major industry players. However, given our resource constraints, we implemented the best feasible solution. We hope this work encourages the community to prioritize data quality and scale our methods further.
>
>
>
> **Q2: Why not directly use GPT-4o as remote sensing (RS) VLMs?**
>
> **A2:** Thank you for this insightful question. While we use GPT-4o for preference dataset judging, this doesn't imply it excels at understanding RS images—many studies demonstrate the GPT series' limitations in RS image understanding [1], [2]. We leverage GPT-4o for judging because we believe that comparing response quality is easier than generating good answers directly.
>
> Moreover, we didn't use RS-VLMs since they have relatively inferior instruction-following capabilities (their largest existing size is 13B, and their fine-tuning datasets usually do not contain data similar to our preference scoring task). Our human verification study also confirms GPT-4o's superiority (92.6% vs. 82.8% for LHRS-Bot-Nova) in judging data quality.
>
> The motivation for building custom RS-VLMs over GPT-4o stems from cost and privicay concerns. Figure 8 demonstrates that ScoreRS outperforms GPT-4o in RL training while being more cost-effective: a single RL run costs \\$ 360 use GPT4o API versus \\$ 256 using ScoreRS on rented cloud compute (16 A100s × 8 hours × \\$2/card / hour).
>
>
>
> [1] Vrsbench: A versatile vision-language benchmark dataset for remote sensing image understanding. Li et. al. NeurIPS 2024
>
> [2] GEOBench-VLM: Benchmarking Vision-Language Models for Geospatial Tasks. Danish et. al. ICCV2025
>
>
>
> **Q3: Using a training scorer with automatic VLMs to filter data for fine-tuning the CLIP model seems circular.**
>
> **A3:** Using a qualifier model to filter training data is actually standard practice for improving model performance. Once trained, the scorer can filter diverse data sources, enabling flagship models to train on smaller, higher-quality datasets. This approach delivers better performance while being more cost-effective through reduced data requirements, faster convergence, and lower computational costs. We believe this trade-off is worthwhile.
>
>
>
> **Q4: What happens when experiments extend beyond the 30% data saving threshold?**
>
> **A4:** Thank you for this question. We tested extreme thresholds of 20% and 10%. Results are shown below:
>
> | Saving Ratio-Method | Classification | Retrieval |
> | :------------------ | :------------: | :-------: |
> | 30%-ScoreRS         |     70.97      |   27.11   |
> | 30%-CLIP-Score      |     69.98      |   26.36   |
> | 20%-ScoreRS         |     68.44      |   27.01   |
> | 20%-CLIP-Score      |     67.70      |   26.69   |
> | 10%-ScoreRS         |     63.19      |   26.16   |
> | 10%-CLIP-Score      |     62.89      |   26.05   |
>
> The results demonstrate that data quantity also remains crucial for maintaining performance, particularly for classification tasks. Extreme ratios significantly degrade classification performance, likely because datasets with numerous classes require sufficient samples per class. However, retrieval tasks are more resilient, as high-quality data effectively captures essential text-to-image alignment concepts, maintaining similar performance even under extreme data reduction.
>
>
>
> **Q5: Suggestions for writing and related work.**
>
> **A5:** Thank you for these valuable suggestions. We will expand the main text to include detailed explanations of our scoring system criteria and enhance the related work section by incorporating additional data building strategies for RS VLMs.

---

> > ### Comment · Reviewer_5qQo · 2025-08-04
> > **Thank you for responding to the concerns**
> >
> > The authors have successfully responded to most of the issues that I felt were weaknesses. Here are a few follow-ups that could strengthen the paper. I would also highly recommend adding these points to the paper for future readers.
> >
> > In response to Q1,A1. A good qualitative experiment that also adds to the paper would be to show how the scorer reacts to captions for an image, which are good/bad on different axes of goodness proposed in the paper. This would partially hand weakness #2  from reviewer CnMa. It can also help in showing that even if the generated caption data is of low quality the scorer trained via RL is good at OOD data for example say from humans.
> >
> >
> > In response to A3, When the authors say, "Using a qualifier model to filter training data is actually standard practice", citations are needed. While I do believe rule-based qualifier models like Datacomp works, this paper is specifically using MLLMs, which are trained on similar/extended data sources over CLIP, which is why I believe it to be circular. This is not the case for rule-based qualifier models.
> >
> > Response from Q4 should be added to the paper.

---

> > > ### Author Response · Authors · 2025-08-04
> > > **Thank you for these valuable suggestions.**
> > >
> > > We truly thank you for your valuable feedback and suggestions for improving our work.
> > >
> > > + Excellent suggestion! We will add more qualitative ranking results from different scoring dimensions based on our proposed scoring system to further validate and unveil the working mechanism of our ScoreRS.
> > >
> > > + Yes, you are absolutely correct. Rule-based data filtering (such as deduplication, grammar checking, and heuristic rules) is the most common and actually the first indispensable step for data filtering. However, as VLM data becomes more complex (such as multi-turn CoT reasoning data) and the definition of "good data" becomes more advanced compared to basic cleanliness and correctness (such as seeking more sophisticated and information-rich data), many advanced VLLMs have adopted model-based methods to score and filter data, including QwenVL [1], InternVL [2], DeepSeek [3], LLaMA [4], and Pixtral [5]. Moreover, using scoring models to filter data is also a standard recipe for building large-scale datasets such as FineWeb-Edu [6].
> > >
> > > [1] Bai, Shuai, et al. "Qwen2. 5-vl technical report." arXiv preprint arXiv:2502.13923 (2025).
> > >
> > > [2] Zhu, Jinguo, et al. "Internvl3: Exploring advanced training and test-time recipes for open-source multimodal models." arXiv preprint arXiv:2504.10479 (2025).
> > >
> > > [3] Wu, Zhiyu, et al. "Deepseek-vl2: Mixture-of-experts vision-language models for advanced multimodal understanding." arXiv preprint arXiv:2412.10302 (2024).
> > >
> > > [4] Dubey, Abhimanyu, et al. "The llama 3 herd of models." arXiv e-prints (2024): arXiv-2407.
> > >
> > > [5] Agrawal, Pravesh, et al. "Pixtral 12B." arXiv preprint arXiv:2410.07073 (2024).
> > >
> > > [6] Penedo, Guilherme, et al. "The fineweb datasets: Decanting the web for the finest text data at scale." Advances in Neural Information Processing Systems 37 (2024): 30811-30849.
> > >
> > > + Thank you for your suggestion. We will add these results to our manuscript!

---

### Official Review · Reviewer_CnMa · 2025-07-05

**Clarity:** 1
**Significance:** 4
**Originality:** 4
**Rating:** 4
**Confidence:** 5

**Summary:**

This paper establishes a novel remote sensing vision-language (VL) data perference dataset, including VL data quality dimensions definition, Image-Caption Preference Dataset curation and Vision Instruction Preference Dataset curation. In addition, a novel VL data quality scoring model ScoreRS is proposed and evaluated in practical applications.

**Questions:**

1. Subfigures in Figure 2 should be enlarged. They are too small to read now.
2. The method introduction is almost entirely in text form, with too few diagrams. This is very unfriendly to the readers. Authors should add more necessary figures. For examples, Vision Instruction Preference Dataset section generally introduces three branches, but the only figure is the very small and coarse diagram in the middle of Figure2. The architecture and training phases of ScoreRS doesn't even have a independent framework figure.
3. The main motivation of this paper is quality assessment, but there is no direct accuracy evaluation of ScoreRS itself. All experiments are actually practical applications of ScoreRS. It is necessary to present the evaluation of pure quality scoring task, including both quantitative and qualitative results. The current experimental content is seriously inconsistent with the motivation of the article.
4. Too many elements for a length-limited conference paper. Perhaps it is better to remain the most crucial parts, e.g. datasets and ScoreRS. Then compile the complete content to another journal version.

**Ethical Concerns:**

["NO or VERY MINOR ethics concerns only"]

**Final Justification:**

4: Borderline accept:

**Limitations:**

yes

**Quality:**

3

**Strengths And Weaknesses:**

Strengths:
1. Insightful and valuable topic
2. Extensive experiments
3. High novelty and application value

Weakness:
1. Poor writing
2. Serious lack of visualization
3. Inconsistent  between motivation and expriment settings

---

> ### Author Rebuttal · Authors · 2025-07-29
>
> **We sincerely appreciate your acknowledgment of our work's innovation, value, and the experimental evidence demonstrating its effectiveness. We recognize that in our effort to present comprehensive and rigorous research, certain details may have made the manuscript challenging to follow, potentially causing confusion for readers. We are grateful for all of your constructive suggestions and provide our item-by-item response below:**
>
>
>
> **Q1: Regarding figure size and additional visualization support for understanding.**
>
> **A1:** Thank you for raising this important concern. We acknowledge that due to the paper's formatting constraints, we consolidated multiple components into a single figure, which inadvertently made Figure 2 too dense and difficult to interpret. We will redesign the figure layout and increase the size of individual subfigures to enhance readability and clarity.
>
> Regarding the ScoreRS training process, while our approach follows standard reward model architecture design (with the exception of our novel 3-stage training strategy), we recognize that additional visual aids would improve the paper's accessibility and completeness. We will therefore add dedicated architectural diagrams and training flow charts to better illustrate the ScoreRS framework and its training phases.
>
>
>
> **Q2: Inconsistency between motivation and experimental settings.**
>
> **A2:** Thank you for this valuable feedback. We respectfully disagree that our motivation and experimental design are inconsistent. Our primary motivation is to develop an automated scoring system to assess the quality of remote sensing vision-language (RSVL) data, which is precisely what ScoreRS accomplishes.
>
> Regarding the evaluation methodology, we believe the most compelling approach to validate data quality assessment is through downstream application performance. By using ScoreRS-filtered data to fine-tune vision-language models and measuring the resulting improvements, we demonstrate the practical effectiveness of our quality scoring system. This application-oriented evaluation directly validates whether our scorer can identify truly high-quality data that enhances model performance.
>
> Additionally, we did include direct evaluation of ScoreRS's scoring capabilities. Specifically, we conducted quantitative assessment using a held-out validation set (Table 6), where ScoreRS achieved 92.91% accuracy in distinguishing high-quality from low-quality data. We also provided qualitative analysis by comparing the highest and lowest ranking samples identified by ScoreRS versus CLIP (Figure 12 in the appendix), demonstrating ScoreRS's superior discrimination ability.
>
> We acknowledge that this direct evaluation could be more prominent in the main paper. We will enhance the presentation of these results to better highlight the intrinsic quality assessment capabilities of ScoreRS alongside the downstream application evaluations.
>
>
>
> **Q3: Suggestions for retaining only partial components of our paper.**
>
> **A3:** Thank you for this suggestion. Our goal is to present a holistic framework addressing the natural workflow: How to build an effective quality assessment system (manual or model-oriented )? If model-oriented, what training approach and data should we use? And finally followed by validation through experiments. Each component serves the logical progression of our manuscript.
>
> However, we acknowledge that extensive content may overwhelm readers. We will restructure the presentation by: (1) emphasizing core workflow and motivation in main sections, (2) streamlining key components, and (3) moving detailed analyses to the appendix. This maintains framework completeness while improving accessibility for conference readers.

---

### Note · Authors · 2025-08-13

Dear Reviewers and Area Chairs,

We are truly grateful for all of your efforts in helping us improve our work. We are encouraged that all reviewers have acknowledged our work's significant contribution to the remote sensing (RS) community, recognized our extensive experiments, and appreciated the novelty of our approach.

To further address the reviewers' concerns, we have clarified our motivation and rationale for choosing the source for preference data generation and demonstrated its effectiveness. While we acknowledge that this process is not perfect, it is highly effective. Through our "data cleaning → reinforcement alignment → test-time scaling" pipeline, we successfully transfer human preferences for high-quality data into model biases that align with human judgment, achieving superior performance.

We recognize that the extensive content and experiments in our work create density that may challenge readers. However, our goal is to build a holistic framework that includes developing an effective scoring model to empower comprehensive applications in RS. To improve readability, we will reformat our paper by moving technical details to the appendix and highlighting the main workflow.

Regarding concerns about comparisons with RS-specific CLIP models for data filtering, we have justified our design choices for using other baselines (as there are no RS-specific CLIP models that support long image-caption pairs and multi-turn conversations, and domain-specific models may not perform as well as OpenAI-CLIP due to suboptimal training data). We have added these comparisons, and the results further demonstrate our model's performance. Importantly, as requirements for high-quality vision-language data become increasingly complex (beyond basic grammar), our scoring model offers versatility across different scenarios—from filtering multi-turn conversations, changed captions, model-based reinforcement training and test-time scaling.

Through this work, we aim to bring greater attention to improving the data quality of RS vision-language datasets. Our contribution provides a comprehensive "score model-centric" paradigm that drives improvements not only in data quality but also in post-training and test-time scaling. We believe our work establishes a strong foundation for validation in larger-scale settings and will inspire further advancements in RS vision-language models.

We hope this addresses all concerns and look forward to contributing to the field.

---

### Decision · Program_Chairs · 2025-09-17

**Decision:**

Accept (poster)

**Comment:**

The reviewers propose an automated scoring-based approach for creating a remote-sensing captioning dataset, where a scoring model is trained to assess the quality of image-caption pairs. Reviewers agree on the significance of the problem and the usefulness of the classifier, as well as the preference data that the model is trained on. All reviewers agree on acceptance following the rebuttal.